# Leveraging Sub-Optimal Data for Human-in-the-Loop Reinforcement Learning

**Calarina Muslimani**
University of Alberta
musliman@ualberta.ca

**Matthew E. Taylor**
University of Alberta
Alberta Machine Intelligence Institute
matthew.e.taylor@ualberta.ca

## Abstract

To create useful reinforcement learning (RL) agents, step zero is to design a suitable reward function that captures the nuances of the task. However, reward engineering can be a difficult and time-consuming process. Instead, human-in-the-loop RL methods hold the promise of learning reward functions from human feedback. Despite recent successes, many of the human-in-the-loop RL methods still require numerous human interactions to learn successful reward functions. To improve the feedback efficiency of human-in-the-loop RL methods (i.e., require less human interaction), this paper introduces Sub-optimal Data Pre-training, SDP, an approach that leverages reward-free, sub-optimal data to improve scalar- and preference-based RL algorithms. In SDP, we start by pseudo-labeling all low-quality data with the minimum environment reward. Through this process, we obtain reward labels to pre-train our reward model *without* requiring human labeling or preferences. This pre-training phase provides the reward model a head start in learning, enabling it to recognize that low-quality transitions should be assigned low rewards. Through extensive experiments with both simulated and human teachers, we find that SDP can at least meet, but often significantly improve, state of the art human-in-the-loop RL performance across a variety of simulated robotic tasks.

## 1 Introduction

In reinforcement learning (RL), an agent's objective is to interact with an environment and maximize its total (discounted) expected reward. The reward hypothesis further maintains that a well-specified reward function is sufficient for an agent to learn to solve a task (Sutton & Barto, 2018). However, defining a reward function that precisely captures all task complexities is often tedious and non-trivial (Booth et al., 2023). There have been notable examples of reward misspecification, in which RL agents discovered and exploited unintended shortcuts in the reward function (Skalse et al., 2022). One notorious example is the CoastRunners game, in which the goal should be to finish a boat race as fast as possible — an RL agent instead gained the most reward by spinning its boat in a circle despite concurrently catching on fire and crashing into other boats (Clark & Amodei, 2016).

A promising alternative is to learn reward functions directly from human feedback. In this paradigm, humans can provide feedback in the form of preferences or scalar signals, which can then be used to learn a reward function that is consistent with human desires (Daniel et al., 2014; Christiano et al., 2017). Despite recent progress, existing preference- and scalar-based RL methods still suffer from high human labeling costs that can require thousands of human queries to learn an adequate reward function (Christiano et al., 2017). Prior work attempts to mitigate this issue through several mechanisms, including active learning (Lee et al., 2021a), data augmentation (Park et al., 2022), semi-supervised learning (Park et al., 2022), and meta-learning (Hejna III & Sadigh, 2023).

Alternatively, our work draws on recent advances in offline RL that have demonstrated the value of low-quality data (Yu et al., 2021). However, its potential in human-in-the-loop RL remains unexplored. As low-quality data is often readily accessible or easy to obtain, this work addresses this gap by asking the question:

Can we leverage sub-optimal, unlabeled data
to improve learning in human-in-the-loop RL methods?

To that end, we present Sub-optimal Data Pre-training, *SDP*, a tool for human-in-the-loop RL algorithms to increase human feedback efficiency. SDP leverages sub-optimal trajectories by pseudo-labeling all transitions with the minimum environment reward. The now pseudo-labeled sub-optimal data serves two purposes. First, we pre-train a regression-based reward model by applying standard supervised learning to minimize the mean squared loss. Intuitively, this pre-training provides the reward model a head start, biasing it towards assigning lower reward values to these low-quality transitions. Second, we initialize the RL agent's replay buffer with the sub-optimal data and make learning updates to the RL agent. This process changes the RL agent's policy and provides different behaviors for the human to provide feedback on (relative to learning with no initial sub-optimal data). This ensures that when the human teacher provides feedback, their time is used efficiently, avoiding redundant feedback on the existing sub-optimal data. Afterward, we follow the standard preference- or scalar-based RL protocol.

This paper's core contribution is showing that we can harness the availability of low-quality, reward-free data for human-in-the-loop RL approaches by pseudo-labeling it with minimum rewards and treating it as a prior for learning reward models. We first validate the utility of SDP in extensive simulated teacher experiments, combining it with four scalar- and preference-based RL algorithms. These experiments show that SDP significantly improves the feedback efficiency in complex tasks from both the DeepMind Control (DMControl) (Tassa et al., 2018) and Meta-World (Yu et al., 2020) suites. Crucially, we further highlight the real-world applicability of SDP by demonstrating its success with human teachers in a 16-person user study. Overall, this work takes an important step toward considering how human-in-the-loop RL approaches can take advantage of readily available sub-optimal data.

## 2 RELATED WORK

**Human-in-the-Loop RL**    Several approaches in human-in-the-loop RL allow agents to leverage human feedback to adapt or learn new behavior. Learning from demonstration is one such methodology that allows a human to provide examples of desired agent behavior (Argall et al., 2009). Human demonstration data has been used to shape the environment's reward function (Brys et al., 2015), develop a reward function from scratch (Abbeel & Ng, 2004), or bias the agent's policy towards certain actions (Taylor et al., 2011). Although demonstrations can be a rich source of feedback, they are often expensive to obtain and may require domain experts (Dragan & Srinivasa, 2012).

Another approach is learning from preference-based feedback where a teacher provides preferences between two or more sets of agent behavior (Christiano et al., 2017). Preference learning has been popularized in recent years as it can require less effort and expertise compared to providing demonstrations. To further reduce the amount of human interaction required, several strategies have been introduced. This has included combining preferences with demonstrations (Ibarz et al., 2018; Bıyık et al., 2022), unsupervised pre-training (Lee et al., 2021a), bi-level optimization (Liu et al., 2022), semi-supervised learning (Park et al., 2022), data augmentation (Park et al., 2022), uncertainty-based exploration (Liang et al., 2022), meta-learning (Hejna III & Sadigh, 2023), and active learning approaches (Hu et al., 2024). Despite its popularity, some argue that comparison feedback might not capture the full intricacies of human preferences, as oftentimes the human is limited to choosing between two options (Daniel et al., 2014; White et al., 2024).

As a result, another body of work focuses on learning from scalar feedback where human teachers can provide scalar signals to evaluate an agent's behavior (Knox & Stone, 2009; Griffith et al., 2013; Loftin et al., 2016; White et al., 2024). Several works use scalar feedback to either learn a reward model (Daniel et al., 2014; Cabi et al., 2020) or an action-value function (Knox & Stone, 2009; 2013; Warnell et al., 2018) via regression.

**Learning from Sub-Optimal Data**    SDP aims to leverage sub-optimal data for scalar- and preference-based RL algorithms. However, learning from low-quality data or negative examples has been applied in other areas of RL and imitation learning (Chen et al., 2021; Tangkaratt et al., 2021). In standard RL, several works use sub-optimal demonstrations to initialize a policy (Taylor et al., 2011; Hester et al., 2018; Gao et al., 2019). In goal-conditioned RL, Hindsight-Experience-Replay

uses failed episodes by treating them as a success with respect to a different goal (Andrychowicz et al., 2017). In inverse reinforcement learning (IRL), Shiarlis et al. (2016) proposed a constrained optimization formulation that can accommodate both successful and failed demonstrations. Brown et al. (2019) makes use of ranked demonstrations to learn a reward function in IRL. Later work in IRL automatically generates ranked trajectories by adding increasing amounts of noise to a learned policy (Brown et al., 2020). Lastly, in offline RL, Singh et al. (2020) leverages sub-optimal transitions from multiple prior tasks and assigns reward labels according to the current task reward function. Our work is most closely related to Yu et al. (2022), which leverages reward-free, sub-optimal data in the offline RL setting by pseudo-labeling all transitions with zero and adding them to the RL agent's replay buffer. However, we found that directly applying this approach to the human-in-the-loop RL setting was ineffective (see Appendix D.5).

## 3 BACKGROUND

In the RL paradigm, agents interact with an environment to maximize the total (discounted) expected reward it can achieve. This interaction is modeled as a Markov Decision Process (MDP) which consists of $\langle \mathcal{S}, \mathcal{A}, T, r, \gamma \rangle$. At every time-step $t$, the agent receives a state $s_t \in \mathcal{S}$ from the environment and chooses an action $a_t \in \mathcal{A}$. The transition function, $T$, determines the probability of transitioning to state $s_{t+1}$ and receiving reward $r_{t+1}$, given the agent was in state $s_t$ and executed action $a_t$. The environment then provides the agent $r_{t+1}$. The agent attempts to learn a policy, $\pi : \mathcal{S} \to \mathcal{A}$, that maximizes the expected return $\mathbb{E}[G] = \sum_{k=0}^{\infty} \gamma^k r_{t+k+1}$, which is defined as the expected sum of discounted future rewards with discount factor $\gamma \in [0, 1)$.

### 3.1 REWARD LEARNING FROM HUMAN FEEDBACK

This paper assumes that we are in a reward-free paradigm, an MDP/R setting, where our goal is to (1) learn a reward function, $\hat{r}$, from human feedback and (2) learn a policy that maximizes the total (discounted) expected $\hat{r}$. We follow the standard reward learning framework that uses supervised learning to learn a parameterized reward function, $\hat{r}_\theta$, with parameters $\theta$ (Christiano et al., 2017). In both scalar- and preference-based settings, we consider trajectory segments $\sigma$, where $\sigma$ consists of a sequence of states and actions: $\{s_t, a_t, s_{t+1}, a_{t+1}, ..., s_{t+k}, a_{t+k}\}$, with $k$ as the segment size.

**Preference-based Reward Learning** In preference-based learning, two segments, $\sigma^0$ and $\sigma^1$, are compared by a teacher, yielding $y \in \{0, 0.5, 1\}$. Specifically, if the teacher preferred segment $\sigma^1$ over segment $\sigma^0$, then $y$ is set to 1, and if the converse is true $y$ is set to 0. If both segments are equally preferred, then $y$ is set to 0.5. As feedback is collected, it is stored as tuples $(\sigma^0, \sigma^1, y)$ in the reward model data set $D_{RM}$. In general, if $\sigma^i > \sigma^j$, then the segment $\sigma^i$ is preferred by the teacher over segment $\sigma^j$. We follow the Bradley-Terry model (Bradley & Terry, 1952) to define a preference predictor using the reward function $\hat{r}_\theta$:

$$P_\theta(\sigma^1 > \sigma^0) = \frac{\exp(\sum_t \hat{r}_\theta(s_t^1, a_t^1))}{\sum_{i \in \{0,1\}} \exp(\sum_t \hat{r}_\theta(s_t^i, a_t^i))} \tag{1}$$

Intuitively, this model assumes that the probability of the teacher preferring a segment depends exponentially on the total sum of predicted rewards along the segment. To train the reward function, we can use supervised learning where the teacher provides the labels $y$. More specifically, we update $\hat{r}_\theta$ by minimizing the standard binary cross-entropy objective:

$$L^{CE}(\theta, D) = -E_{(\sigma^0, \sigma^1, y) \sim D}\left[(1 - y)\log P_\theta(\sigma^0 > \sigma^1) + y \log P_\theta(\sigma^1 > \sigma^0)\right] \tag{2}$$

**Scalar-based Reward Learning** The primary difference between scalar and preference-based reward learning is that in scalar-based learning, the human teacher assigns numerical ratings to trajectory segments. In this setting, the comparisons between segments are implicit. More concretely, a teacher assigns a scalar value $y$ to a segment $\sigma^i$, and as feedback is collected, it is stored as tuples $(\sigma^i, y)$ in the reward model data set $D_{RM}$. We then apply standard regression and update $\hat{r}_\theta$ by minimizing the mean squared error:

$$L^{MSE}(\theta, D) = E_{(\sigma^i, y) \sim D}\left[(y - \sum_t \hat{r}_\theta(s_t^i, a_t^i))^2\right] \tag{3}$$

**Human Studies in Human in the Loop RL**  To evaluate human-in-the-loop RL algorithms, a common protocol is the use of simulated teachers, where feedback is provided according to a ground truth reward function. This can be useful, as it offers an efficient and controlled evaluation setting. However, we argue that it is essential to evaluate human-in-the-loop RL algorithms with *real human feedback*. This is especially important in light of recent work finding discrepancies between preference learning algorithms when using simulated teacher feedback versus human feedback (Metcalf et al., 2024).

## 4  SUB-OPTIMAL DATA PRE-TRAINING

In this section, we present SDP, a tool that leverages sub-optimal trajectories to improve the feedback efficiency for human-in-the-loop RL. We refer to sub-optimal trajectories as sequences of $(s, a)$ pairs such that:

$$r(s_t, a_t) - r_{\min} < \epsilon \quad \forall t \in [t, t + k] \quad \text{for some small} \quad \epsilon > 0, \tag{4}$$

where $k$ is the segment size and $r_{\min}$ is the minimum possible environment reward. Equation 4 essentially expresses that the rewards achieved along a sub-optimal trajectory should be close to $r_{\min}$. However, it is important to note that in practice we do not have access to the ground truth reward. This prevents us from directly identifying sub-optimal trajectories using Equation 4. Instead, we rely on selecting trajectories that we estimate will align with this criterion, such as gathering trajectories via a random policy.

Once sub-optimal trajectories are collected, we take the approach of pseudo-labeling all transitions with $r_{\min}$. The goal of SDP is then to use this pseudo-labeled data to create a prior for reward models in human-in-the-loop RL methods (see Figure 1). Its simplicity enables SDP to be used in conjunction with any off-the-shelf human-in-the-loop RL algorithm that learns a reward function from human feedback.

SDP comprises two phases: (1) the reward model pre-training phase and (2) the agent update phase. In the reward model pre-training phase, we first gather a data set, $D_{\text{sub}}$, of $N$ sub-optimal state, action transitions. We then pseudo-label all transitions in $D_{\text{sub}}$ with rewards of $r_{\min}$, resulting in $D_{\text{sub}} = \{s_i, a_i, r_{\min}\}_{i=1}^N$. $D_{\text{sub}}$ is then used to optimize the reward model $\hat{r}_\theta$ with the mean squared loss in Equation 3. As a result, the reward model $\hat{r}_\theta$ learns to associate all sub-optimal transitions with a low reward. Without such a prior, the reward model would initially have random estimates for these transitions; while the only way to improve such estimates is to obtain feedback from a teacher. Therefore, the reward model pre-training phase provides a valuable reward initialization without requiring any human feedback.

Next, in the agent update phase, we initialize the RL agent's replay buffer $D_{\text{agent}}$ with $D_{\text{sub}}$. The RL agent then briefly interacts with its environment and performs gradient updates according to its loss functions. The agent update process changes the RL agent's policy and generates new transitions, which are then stored in both the agent's replay buffer $D_{\text{agent}}$ and the reward model's data set $D_{\text{RM}}$. It is important to note that in standard scalar- and preference-based reward learning, we query the teacher for feedback on trajectory segments sampled from $D_{\text{RM}}$. Therefore, adding new transitions into $D_{\text{RM}}$ during the agent update phase is necessary to ensure that the teacher does not provide redundant feedback to the original sub-optimal transitions (as $D_{\text{RM}}$ was empty prior to the agent update phase). When it is time for the teacher to provide their first set of feedback, the feedback can cover a different region of the state and action space, relative to the original sub-optimal data. In Appendix D.1, Figure 8 we empirically show that the agent update phase changes the RL agent's policy by performing policy rollouts and analyzing the differences in state distributions. See Algorithm 1 for the complete pseudocode.

At first glance, labeling sub-optimal transitions with an incorrect reward may seem problematic, as incorrect labels do introduce statistical bias into the reward model and the RL agent's value network. However, as the transitions are sub-optimal, we observe that the bias for using an incorrect reward is low (see Figure 9 in Appendix D.2). Moreover, by using the sub-optimal transitions, we increase the overall amount of data used by both models, which can *decrease* the models' variance, as shown in the offline RL setting (Yu et al., 2022).

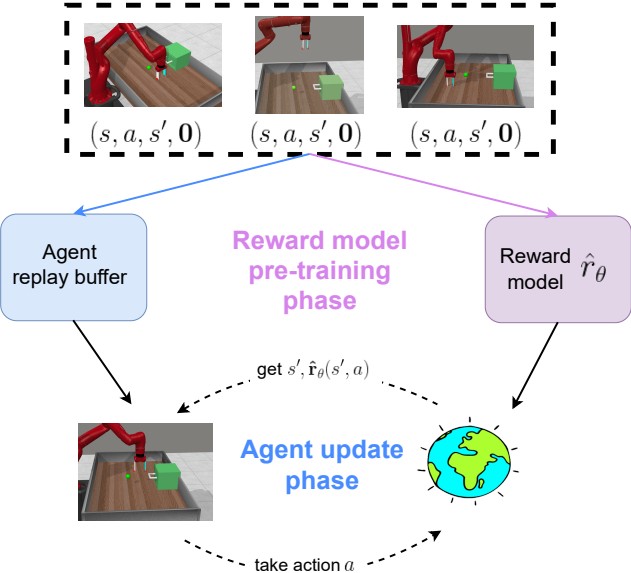

Figure 1: Overview of SDP: After obtaining a data set of sub-optimal trajectories, we pseudo-label the transitions with rewards of $r_{\min}$ (e.g., $r_{\min} = 0$). We then pre-train the reward model $\hat{r}_\theta$ using this data set. During the agent update phase, we initialize the RL agent's replay buffer with the same pseudo-labeled data set. The agent then interacts in the environment and makes learning updates to obtain new behaviors for a teacher to give feedback.

---

**Algorithm 1** SDP

---

**Require**: Reward model $\hat{r}_\theta \leftarrow \theta$ randomly initialized, Reward model data set $D_{\text{RM}} \leftarrow \emptyset$, RL agent with replay buffer $D_{\text{agent}} \leftarrow \emptyset$, Sub-optimal data set $D_{\text{sub}}$ with reward labels $r_{\min}$

 1: // REWARD MODEL PRE-TRAIN PHASE
 2: **for** each gradient step **do**
 3:         Optimize $\hat{r}_\theta$ on $D_{\text{sub}}$ with $L^{MSE}$ (Equation 3)
 4: **end for**
 5: // AGENT UPDATE PHASE
 6: $D_{\text{agent}} \leftarrow D_{\text{sub}}$
 7: **for** each time-step $t$ **do**
 8:         Collect $s_{t+1}$ by taking action $a_t \sim \pi(s_t)$
 9:         Store $(s_t, a_t, \hat{r}_\theta, s_{t+1})$ in $D_{\text{agent}}$
10:         Store $(s_t, a_t)$ in $D_{\text{RM}}$
11:         Update RL agent with $D_{\text{agent}}$
12: **end for**
13: Begin scalar- or preference-based RL using pre-trained $\hat{r}_\theta$, RL agent, and $D_{\text{agent}}$, $D_{\text{RM}}$

---

## 5 EXPERIMENTS

This section considers the following four research questions (RQ's):

RQ 1: Can SDP improve upon existing scalar- and preference-based RL methods?

RQ 2: Can SDP effectively leverage sub-optimal trajectories from different tasks to improve performance on a target task?

RQ 3: Can SDP be used with real human feedback?

RQ 4: How sensitive is SDP to various hyperparameters?

## 5.1 EXPERIMENTAL DESIGN

To demonstrate the versatility of SDP, we apply SDP to both preference and scalar-based RL approaches. However, as preference feedback can be less time-consuming than scalar feedback, we primarily concentrate on preference-based RL in our experiments, exploring scalar feedback in a smaller capacity. For the preference-based experiments, we combine SDP with four contemporary preference-based algorithms: PEBBLE (Lee et al., 2021a), RUNE (Liang et al., 2022), SURF (Park et al., 2022), and MRN (Liu et al., 2022). We benchmark the performance of the four algorithms augmented with SDP against their original versions without SDP, as well as against SAC. We treat SAC (Haarnoja et al., 2018) as an oracle (i.e., upper bound) because it learns while accessing the ground truth reward function, which is unavailable to the other algorithms. For the scalar-based experiments, we combine SDP with R-PEBBLE (a regression variant of PEBBLE). We compare SDP + R-PEBBLE against R-PEBBLE, Deep TAMER (Warnell et al., 2018) (a scalar feedback RL algorithm), and SAC. We note that SAC is the core RL algorithm used across all baselines.

**Implementation Details**   For SDP, we collected sub-optimal trajectories via a random policy. In particular, we used $50000$ state, action transitions for all experiments in Section 5.2. Note that we do not require explicit access to a sub-optimal policy; we only require state, action transitions from said policy. Moreover, to ensure a fair comparison across algorithms, we maintained equal feedback budgets for all algorithms within each environment, while adjusting the budget across environments to reflect their degree of difficulty. See Appendix A for a complete overview of the implementation process and specific hyperparameters for all algorithms.

**Evaluation**   We show average offline performance (i.e., freeze the policy and evaluate it with no exploration) over ten episodes using either the ground truth reward function (DMControl experiments) or the success rate (Meta-World experiments). It is important to note that only SAC has access to the ground truth reward function. We perform this evaluation every $10000$ training steps. To systemically evaluate performance, we use a simulated teacher that provides either a scalar rating of a single trajectory segment or preferences between two trajectory segments according to the ground truth reward function. To thoroughly test the effectiveness of SDP, we perform evaluations on four robotic locomotion tasks from the DMControl Suite: Walker-walk, Cheetah-run, Quadruped-walk, and Cartpole-swingup, and five robotic manipulation tasks from Meta-World: Hammer, Door-unlock, Door-lock, Drawer-open, and Window-open. In our experiments, the results are averaged over five seeds with shaded regions or error bars indicating $95\%$ confidence intervals. To test for significant differences in final performance (i.e., the undiscounted return) and learning efficiency (i.e., the total area under the return curve, AUC), we perform Welch $t$-tests (equal variances not assumed) with a p-value of $0.05$. See Appendices D.8 and D.9, Tables 9-14 for a summary of final performance and AUC across all experiments.

## 5.2 LOCOMOTION AND MANIPULATION RESULTS

**Preference Feedback Experiments**   We first address RQ 1 by evaluating the utility of SDP in the preference-based RL setting. Considering all four preference-based algorithms in the nine environments, SDP significantly ($p < 0.05$) improved learning (i.e., either final performance or AUC) in 23 out of the 36 experiments (see Figure 2). In the remaining experiments, there were no significant differences in the performance between SDP and the baseline algorithms. The addition of SDP (i.e., SDP + base algorithm) *never* statistically hurt performance.

**Scalar Feedback Experiments**   Continuing our investigation into RQ 1, we now evaluate the performance of SDP in the scalar-based RL setting. We performed evaluations in Walker-walk, Cheetah-run, and Quadruped-walk. In Figure 3, we found that SDP (purple curve) significantly improves either the final performance or AUC compared to R-PEBBLE (navy curve) and Deep TAMER (yellow curve). More impressively, we found that SDP achieves comparable final performance to SAC (red curve), which uses the ground truth reward function, using as little as 60 feedback queries.

**Leveraging Different Task Data in SDP**   Our previous experiments showed that SDP can leverage sub-optimal data from the target task to improve human-in-the-loop RL methods. Now, addressing RQ 2, we investigate whether SDP can similarly use sub-optimal data from *related tasks* to

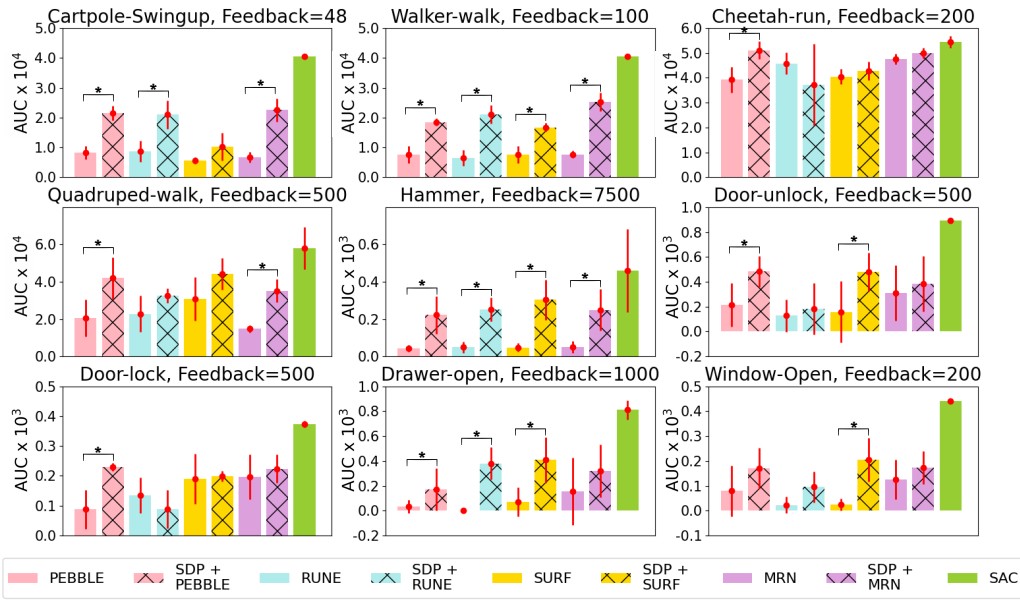

Figure 2: Results from the preference feedback experiments in the DMControl and Meta-World suites show mean AUC ± 95% confidence intervals. * indicates that SDP + the base preference learning algorithm achieves a statistically greater score than the base preference learning algorithm.

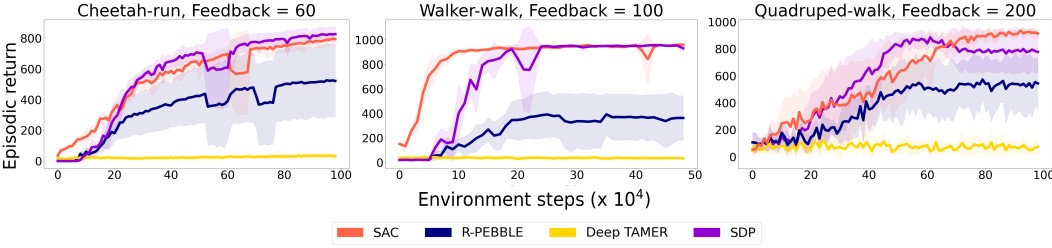

Figure 3: In the scalar feedback experiments on the DMControl environments, SDP significantly outperforms R-PEBBLE and Deep TAMER and achieves comparable performance to SAC.

improve performance on the target task. We perform three preference learning experiments, comparing PEBBLE with SDP + PEBBLE, using sub-optimal data from a different prior task that has the same virtual robot: (1) Walker-stand for Walker-walk, (2) Quadruped-walk for Quadruped-run, and (3) Drawer-open for Door-open. To obtain the sub-optimal data for the prior tasks, we gathered transitions from partially trained policies as opposed to using random policies. Each partially trained policy achieved a final score of approximately 15-20% of that achieved by a fully trained policy. This ensured that the distribution of sub-optimal data differed between the prior and target tasks. See Appendix A.2 for further details on the experiment setup. Figure 4 demonstrates that in all three tested environments, SDP can successfully leverage sub-optimal data from related tasks (green curve) as it achieved similar performance to SDP when leveraging target task data (purple curve).

## 5.3 Preference Learning with Human Feedback

To evaluate human-in-the-loop RL algorithms, we argue that it is critical to understand their efficacy with human teachers. However, human user studies do not appear to be widely adopted in the current literature. To empirically investigate this, we conducted a survey of 45 preference learning studies

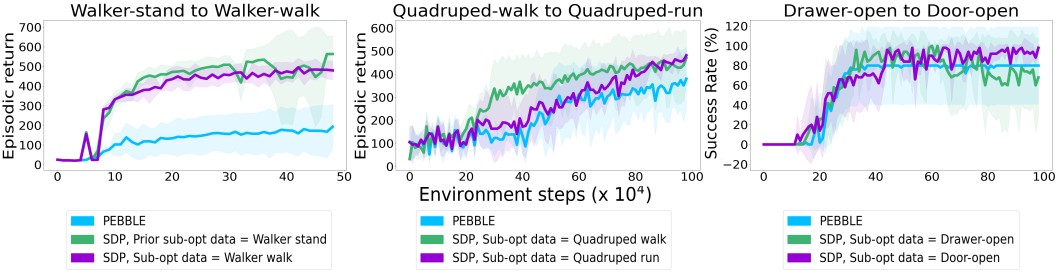

Figure 4: This highlights that SDP can leverage sub-optimal data from different prior tasks as it performed comparable to SDP when using target task sub-optimal data.

from 2012 to 2024[1] to understand the prevalence of human user studies in the preference learning literature. We found that *fewer than* 50% tested their proposed algorithms with human participants who were not also the authors. Moreover, of those studies that did involve human participants, only 41% included non-expert individuals, while none provided sufficient demographic information about their participants (e.g., gender, race/ethnicity, level of education). These limitations raise significant concerns about whether and how current preference learning algorithms generalize to different populations.

To that end, we now address RQ 3 and evaluate the efficacy of SDP with real human feedback. We conduct an ethics-approved human-subject study of 16 participants (9 male, 7 female). Age ranges were collected: 18-24 (6), 25-30 (7), and 50-70 (3). Participants self-identified their membership in racial groups: South Asian (6), East Asian (3), White (4), Middle Eastern or North African (2), and multi-racial (1). The participants' highest educational attainments were: high school diploma (3), Bachelor's degrees (8), and Master's degrees (5), and their expertise varied across AI/ML (12), other computer science topics (1), and non-computer science fields (3). This highlights the diversity of our participants in terms of demographics and expertise.

This user study compares the performance of SDP and PEBBLE in two DMControl environments: Pendulum-swingup (with 7 participants) and Cartpole-swingup (with 12 participants). We focused our comparison to PEBBLE as it performed comparably to the other preference learning baselines in Section 5.2. We use a between-subjects experimental design — each participant provides preferences for a single seed of both SDP and PEBBLE algorithms. The preference budgets for Pendulum-swingup and Cartpole-swingup were 40 and 48, respectively. We selected these environments specifically because they could be solved with fewer preferences, aiming to reduce the overall time commitment required from participants. Each trial for a single environment lasted approximately 1–1.5 hours. See Appendix B for more details including user instructions and interface.

We visualize both algorithms' final performance and AUC in Figure 5. We found that in both environments SDP (purple plots) maintains significant (p < 0.05) performance gains over PEBBLE (blue plots) in terms of either final performance or AUC. In Cartpole-swingup, we also observed consistent performance from SDP regardless of whether the teacher was human or simulated. This suggests that our prior results with simulated teachers can generalize to settings where human feedback is provided. Moreover, we observed no significant differences in SDP's effectiveness across demographic factors, including gender, age, educational, and computer science background (see Appendix C, Tables 7 and 8). This finding is particularly encouraging for the potential real-world deployment of SDP, highlighting its usability for non-expert users across diverse demographics.

## 5.4 ABLATION AND SENSITIVITY STUDIES

To further understand the effectiveness of SDP, we perform further analysis of SDP across three dimensions: (1) the phases of SDP, (2) the number of feedback queries, and (3) the amount of sub-optimal data. This analysis aims to address RQ 4 and provide a deeper understanding of the factors influencing SDP's performance. For these experiments, we focus on SDP + R-PEBBLE in the Walker-walk environment. Additional results for Cheetah-run can be found in Appendix D.5.

---

[1]See Appendix B.2 for more details.

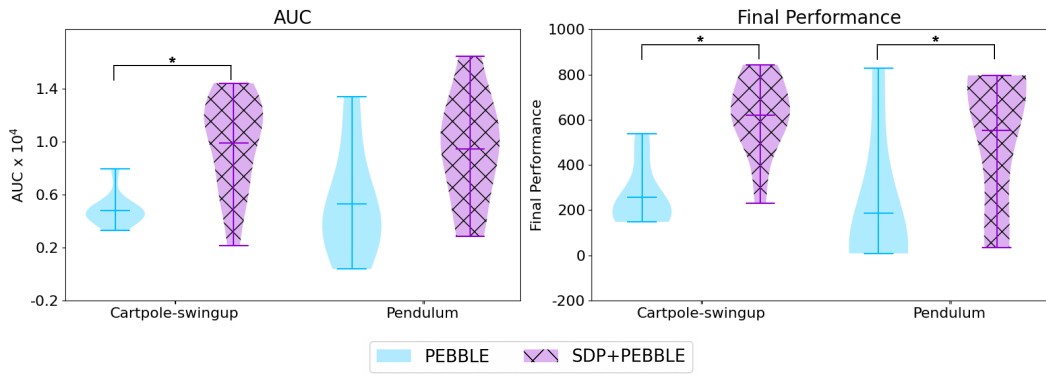

Figure 5: This demonstrates that SDP can significantly outperform PEBBLE in terms of both AUC (left) and final performance (right) even when human teachers are providing preferences. * denotes a statistically significant difference between SDP and PEBBLE.

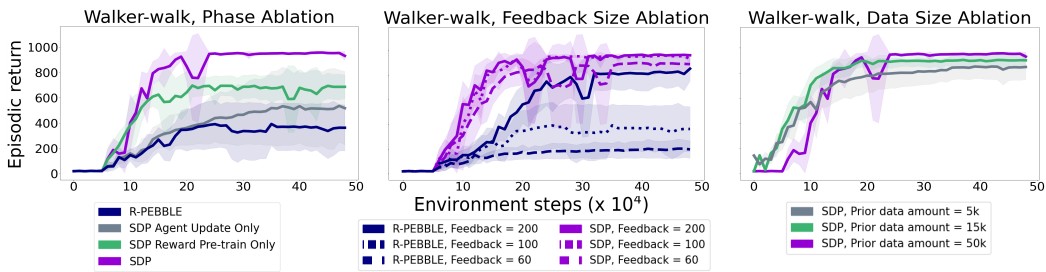

Figure 6: These figures show ablation and sensitivity studies of SDP in Walker-walk.

**SDP Component Analysis**  First, we evaluate the effect of each phase of SDP individually, the reward model pre-train phase, and the agent update phase. Figure 6 (leftmost) demonstrates the importance of using both phases in SDP for scalar-based RL approaches. We found that the SDP variants that only use one of the phases (green and gray curves) result in worse performance than the full SDP (purple curve).

**Effect of Feedback Amount**  We evaluate SDP and R-PEBBLE with feedback budgets $\in [60, 100, 200]$ to analyze the impact of feedback quantity on performance. As shown in Figure 6 (middle), SDP (purple curves) consistently outperforms R-PEBBLE (navy curves), further demonstrating its effectiveness across varying feedback levels.

**Effect of Sub-Optimal Data Amount**  We evaluate the performance of SDP using varying amounts of sub-optimal transitions $\in [5000, 15000, 50000]$. Figure 6 (rightmost) indicates that while 5000 transitions (gray curve) led to the poorest performance, increasing this amount to 15000 or 50000 (green and purple curves) yielded comparable or improved results, suggesting that more sub-optimal data can benefit SDP.

## 6  CONCLUSION

In this work, we present SDP, an approach that improves the feedback efficiency for human-in-the-loop RL algorithms. SDP is specifically designed to leverage reward-free, sub-optimal data for scalar- and preference-based RL approaches. By pseudo-labeling low-quality data with the minimum environment rewards, we can pre-train the reward model without the need for human labeling. This provides the reward model with a head start in learning. This head start allows the reward model to learn to associate low-quality transitions with low reward values, even before receiving any actual

human feedback. Our simulated teacher experiments in DMControl and Meta-World suites demonstrate that SDP can significantly improve a variety of preference- and scalar-based reward learning algorithms. Importantly, we further validate the real-world applicability of SDP by demonstrating its success in a 16-person user study. This work takes an important step towards considering how sub-optimal data can be leveraged for human-in-the-loop RL.

## ACKNOWLEDGMENTS

Part of this work has taken place in the Intelligent Robot Learning Lab at the University of Alberta, which is supported in part by research grants from Alberta Innovates; Alberta Machine Intelligence Institute (Amii); a Canada CIFAR AI Chair, Amii; Digital Research Alliance of Canada; Mitacs; and the National Science and Engineering Research Council.

## ETHICS STATEMENT

The goal of preference and scalar-based RL is to learn reward functions that encode human preferences. However, this necessitates interaction with human users, raising concerns about potential biases in the collected preferences. If these preferences are primarily drawn from a non-representative group, the resulting RL system may unfairly prioritize the desires or needs of that group over others. To that end, we performed a human subject study that was reviewed and approved by an external ethics committee. This ensured the responsible and ethical collection of human preferences. In addition, we took care to gather participants from a wide range of both demographic and educational backgrounds, which we detail in Section 5.3.

## REPRODUCIBILITY STATEMENT

To ensure the reproducibility of our work, we provide a link to our code repository: `https://github.com/cmuslima/SDP_ICLR`. We also provide pseudocode in Section 4. Moreover, we outline extensive training and evaluation details in Section 5.1 and Appendix A.2. We also outline all hyperparameters used for all algorithms and environments in Appendix A.2, Tables $1-5$. Lastly, for the human subject study, we provide details on the instructions provided to the participants in Appendix B as well as a description of the preference interface used. In our released codebase, we also provide the specific code files used to run the experiments in the user study. In addition, an anonymized version of the human subject data will be released, as stated in our ethics approval.

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

# APPENDIX

## A SIMULATED EXPERIMENT DETAILS

### A.1 BENCHMARKS

**PEBBLE** PEBBLE has two primary components: unsupervised exploration and off-policy learning with relabeling. The purpose of the unsupervised exploration phase is to collect diverse experiences for a human teacher to provide feedback on. More specifically, PEBBLE optimizes state entropy to explore the environment. Furthermore, PEBBLE uses off-policy reinforcement learning to learn a policy. PEBBLE specifically uses an off-policy RL algorithm as they are more sample efficient compared to their on-policy counterparts. Then as the reward model changes, PEBBLE relabels all transitions in the RL agent's replay buffer with the latest reward model. This is integral as the reward model is non-stationary, and relabeling the transitions stabilizes the learning process.

To adapt PEBBLE to the scalar feedback setting, we make one minor change to the reward model. In the scalar feedback setting, we use a scripted teacher that provides a scalar rating of a single trajectory segment. Therefore, the only update to PEBBLE is with respect to the loss function. Instead of using the cross-entropy loss in Equation 2, we use the mean-squared error loss in Equation 3.

**RUNE** RUNE is a preference learning algorithm (built on top of PEBBLE) that uses an uncertainty-based exploration strategy to improve feedback efficiency. To encourage exploration for the SAC agent, RUNE adds an intrinsic reward component based on the standard deviation in the reward model.

**SURF** SURF is another preference learning algorithm (built on top of PEBBLE) that improves feedback efficiency by using semi-supervised and data augmentation approaches. To incorporate semi-supervised learning, SURF generates pseudo-labels for unlabeled trajectories by querying the learned reward model. If the reward model confidently (e.g., low output standard deviation) predicts the pseudo-label, then the trajectory, label pair is added to the reward model training data set. Further, SURF proposes a new data augmentation technique that crops sub-sequences of trajectories.

**MRN** MRN is a preference learning algorithm also integrated with PEBBLE. Unlike other PbRL algorithms, MRN is a bi-level optimization algorithm in which the actor and critic are updated in the inner loop, and the reward model is updated in the outer loop. Importantly, the reward model takes into account the performance of the critic on the preference data.

**Deep TAMER** In our scalar feedback experiments, we also consider the Deep TAMER benchmark. In Deep TAMER, scalar feedback is used to learn a human reward function via regression. Then the agent acts greedily with respect to this reward function. Furthermore, the original implementation of Deep TAMER was built on top of DQN. Therefore, there was no separate actor-network. In addition, Deep TAMER only used discrete feedback $\in [-1, 0, 1]$.

To make Deep TAMER a fair benchmark, we made a few adjustments. To start, we allow Deep TAMER to learn from real-valued feedback as done in the other scalar-based experiments. However, instead of using the ground truth reward function as feedback, we use the state-action values from a fully-trained SAC agent. We do this because, in TAMER (and Deep TAMER), the teacher is intended to provide feedback representative of the return. Secondly, in Deep TAMER, feedback is provided per (state, action) pair. Therefore, to make sure Deep TAMER received the same amount of feedback as the other baselines, we used *trajectory segment size × feedback amount* for Deep TAMER only. The other benchmarks receive a scalar feedback value that is the sum of rewards along a trajectory segment. Third, to learn the reward model we use standard regression as described in Section 3.1. Lastly, as our testing environments are continuous state and action, we learn a separate actor policy, similarly done in (Vien et al., 2013).

## A.2 Training Details

In all of our experiments, we use the hyperparameters in Table 1 for the reward models used in all benchmarks. For the agent update phase of SDP, an additional hyperparameter is associated with the number of environment interactions made before the standard preference/scalar feedback learning loop begins. However, for simplicity, we kept the same value as the existing feedback frequency hyperparameter, as feedback frequency also dictates the number of environment interactions made between feedback sessions

Furthermore, we use most of the existing reward model hyperparameters used in PEBBLE, however, we adjusted the following four hyperparameters: feedback frequency, amount of feedback per session, trajectory segment size (only for Meta-world), and activation function for the final NN layer. We adjusted the first two hyperparameters because PEBBLE originally used a significantly larger feedback budget, therefore we wanted the feedback schedule to better reflect a smaller feedback budget. We used a different trajectory segment size for Meta-world because we wanted to keep the segment sizes the same across both the DMControl and Meta-world environments. Moreover, we found that the output activation function could significantly affect learning, therefore we tested all benchmarks using both Tanh (original activation used) and Leaky-ReLU and chose the reward model that achieved better final performance. For the RUNE and SURF baselines, we use any hyperparameters associated with their specific algorithm according to the original paper (see Table 2). For a fair comparison with SDP, we provide all human-in-the-loop baselines (e.g., PEBBLE, R-PEBBLE, Deep TAMER, RUNE, and SURF) with the sub-optimal data set to be used in both the reward model and by the RL agent.

Furthermore, to select trajectory segments for the teacher to provide feedback on, we use uniform sampling in the DMControl tasks and disagreement sampling in the Meta-world tasks. Disagreement sampling is a popular active learning approach in which trajectories with higher uncertainty (based on an ensemble of neural networks) are more likely to be sampled (Christiano et al., 2017). As for the SAC hyperparameters, we use the values found in Tables 3-4.

For the experiments in which we leveraged sub-optimal data from a different task (i.e., Walker-stand, Quadruped-walk, Drawer-open), we gathered 50,000 transitions from partially trained policies. We note that for these experiments, we purposely did not use transitions gathered from a random policy. In these experiments, the prior and target tasks were environments in which the simulated robot was identical. The only difference is the environmental reward. Therefore, the random policy for both environments would be the same. To truly demonstrate transfer, we wanted to ensure we obtained low-quality transitions of the prior task that were different from the target task.

Each partially trained policy achieved a final score of approximately 15-20% of that achieved by a fully trained policy. More specifically, we used the following procedure to train the SAC policies. First, for Walker-stand, we trained a SAC policy for 5,000 time steps, and the average final performance was approximately 194. Second, for Quadruped-walk, we trained a SAC policy for 100,000 timesteps, and the average final performance was approximately 184. Lastly, for Drawer-open, we trained a SAC policy for 50,000 time steps, and the average final success rate was approximately 14%.

| HYPERPARAMETER | VALUE |
|---|---|
| SEGMENT SIZE | 50 |
| RANDOM STEPS (I.E., SUB-OPTIMAL DATA TRANSITIONS) | 50000 |
| UNSUPERVISED EXPLORATION STEPS | 9000 |
| CONSIDER UNSUPERVISED EXPLORATIONS AS SUB-OPTIMAL | FALSE (WALKER-WALK, CARTPOLE-SWINGUP, QUADRUPED-WALK – PBRL) TRUE (OTHERS) |
| FREQUENCY OF FEEDBACK | 20000 (DMCONTROL) 10000 (WINDOW-OPEN, DOOR-UNLOCK, DOOR-LOCK) 5000 (HAMMER, DRAWER-OPEN) |
| FEEDBACK BUDGETS IN ABLATIONS | 2000 (DOOR-OPEN) 1000 (CHEETAH-RUN, WALKER-WALK) |
| FEEDBACK QUERIES PER SESSION | 50 (HAMMER, DRAWER-OPEN) 8 (CARTPOLE-SWINGUP) 20 (OTHERS) |
| SAMPLING SCHEME | DISAGREEMENT SAMPLING (METAWORLD) UNIFORM SAMPLING (DMCONTROL) |
| TRAINING EPOCHS | 200 (WINDOW-OPEN, SURF AND SDP + SURF) 50 (OTHERS) |
| LEARNING RATE | $3 \times 10^{-4}$ |
| INTERMEDIATE NEURAL NETWORK ACTIVATION | LEAKY RELU |
| BATCH SIZE | 128 |
| HIDDEN LAYERS | 4 |
| NEURONS PER HIDDEN LAYER | 128 |
| LOSS FUNCTION | MEAN SQUARED ERROR (SCALAR FEEDBACK) CROSS ENTROPY LOSS (PREFERENCE FEEDBACK) |
| OPTIMIZER | ADAM |
| *For Reward Model Pre-training Phase in SDP* LAST LAYER NEURAL NETWORK ACTIVATION | TANH |
| *For Human-in-the-Loop Algorithm* LAST LAYER NEURAL NETWORK ACTIVATION | LEAKY RELU (PEBBLE, RUNE, SURF) TANH (SDP – HAMMER, DRAWER-OPEN, WALKER-WALK) LEAKY RELU (SDP – OTHERS) |

Table 1: Hyperparameters for the reward model used in all experiments (both preference and scalar feedback variants).

| HYPERPARAMETER | VALUE |
|---|---|
| *Specific RUNE Hyperparameters* | |
| BETA SCHEDULE | LINEAR DECAY |
| BETA INIT | 0.05 |
| BETA DECAY | $10^{-5}$ |
| | |
| *Specific SURF Hyperparameters* | |
| THRESHOLD $\lambda$ | 0.99 |
| UNLABELED BATCH RATIO | 4 |
| LOSS WEIGHT | 1 |
| MIN/MAX LENGTH OF CROPPED SEGMENT | [45, 55] |
| SEGMENT LENGTH BEFORE CROPPING | 60 |
| | |
| *Specific MRN Hyperparameters* | |
| META STEPS | 1000 (WALKER-WALK) |
| | 3000 (QUADRUPED-WALK) |
| | 10000 (DOOR-OPEN) |
| | 5000 (OTHERS) |

Table 2: Baseline Specific Hyperparameters.

| HYPERPARAMETER | VALUE |
|---|---|
| OPTIMIZER | ADAM (Kingma & Ba, 2015) |
| DISCOUNT | 0.99 |
| ALPHA LEARNING RATE | $10^{-4}$ |
| ACTOR BETAS | 0.9, 0.999 |
| CRITIC BETAS | 0.9, 0.999 |
| ALPHA BETAS | 0.9, 0.999 |
| TARGET SMOOTHING COEFFICIENT | 0.005 |
| ACTOR UPDATE FREQUENCY | 1 |
| CRITIC TARGET UPDATE FREQUENCY | 2 |
| INIT TEMPERATURE | 0.1 |
| NETWORK TYPE | MLP |
| NONLINEARITY | ReLU |
| GRADIENT UPDATES PER STEP | 1 |

Table 3: Hyperparameters for SAC that were shared by all algorithms. SAC is the standard RL algorithm that learns from the environment's true reward signal, not via preference learning.

| HYPERPARAMETER | VALUE |
|---|---|
| *DMControl* | |
| BATCH SIZE | 512 (CARTPOLE-SWINGUP) |
| | 1024 (OTHERS) |
| | |
| HIDDEN LAYERS | 2 |
| NEURONS PER HIDDEN LAYER | 256 (CARTPOLE-SWINGUP) |
| | 1024 (OTHERS) |
| | |
| ACTOR/CRITIC LEARNING RATE | $5 \times 10^{-5}$ (CHEETAH-RUN PREFERENCE FEEDBACK) |
| | $10^{-4}$ (CHEETAH-RUN SCALAR FEEDBACK) |
| | $5 \times 10^{-4}$ (WALKER-WALK) |
| | $10^{-4}$ (OTHERS) |
| | |
| TRAINING STEPS | $0.5 \times 10^6$ (WALKER-WALK, CARTPOLE-SWINGUP) |
| | $10^6$ (OTHERS) |
| | |
| *Metaworld* | |
| BATCH SIZE | 512 |
| HIDDEN LAYERS | 3 |
| NEURONS PER HIDDEN LAYER | 256 |
| ACTOR/CRITIC LEARNING RATE | $3 \times 10^{-4}$ |
| | |
| TRAINING STEPS | $0.5 \times 10^6$ (DOOR-LOCK AND WINDOW-OPEN) |
| | $10^6$ (OTHERS) |

Table 4: Specific SAC hyperparameters that were tuned for the DMControl and Metaworld experiments. The majority of these hyperparameters were selected from the PEBBLE repo (Lee et al., 2021a).

| HYPERPARAMETER | VALUE |
|---|---|
| *SAC* | |
| TRAINING STEPS | $0.3 \times 10^6$ |
| BATCH SIZE | 1024 |
| HIDDEN LAYERS | 2 |
| NEURONS PER HIDDEN LAYER | 1024 |
| ACTOR/CRITIC LEARNING RATE | $10^{-4}$ |
| | |
| *Reward Model* | |
| CONSIDER UNSUPERVISED EXPLORATIONS AS SUB-OPTIMAL | FALSE |
| FREQUENCY OF FEEDBACK | 20000 |
| | |
| FEEDBACK QUERIES PER SESSION | 10 (PENDULUM-SWINGUP) |
| | 8 (CARTPOLE-SWINGUP) |
| | |
| SAMPLING SCHEME | UNIFORM SAMPLING |
| BATCH SIZE | 128 |
| LAST LAYER NEURAL NETWORK ACTIVATION | LEAKY RELU |

Table 5: Tuned hyperparameters for human feedback experiments.

## B    HUMAN SUBJECT STUDY DETAILS

Our study was approved by an external ethics committee. In total, 16 users participated in our study, 3 of which participated in both Pendulum-swingup and Cartpole-swingup tasks. As for the amount of participation per task, 7 users provided preferences in Pendulum-swingup and 12 users provided preferences in Cartpole-swingup. During the timeline of our user study, we initially had participants interacting in Pendulum-swingup, however we found that it was very simple, therefore we decided to include Cartpole-swingup, as a more difficult task. Each participant provided preferences for a single run of the environment, as each session took approximately 1 to 1.5 hours. Table 5 shows the specific hyperparameters used.

### B.1    PROTOCOL STEPS

The study took place in person, where the participant was first provided the consent from to review and sign. Then, participants and the researcher, together, reviewed the instructions outlining the objective of the study. The instructions had two components. First, they outlined participant's goal in the study. Second, as done in Christiano et al. (2017), we provided a guide on what constitutes good and bad behavior in each domain.

**Pendulum-swingup**    The first set of instructions for the study are as follows:

1. Objective: for the pole to swing up and balance.
2. Agent controls how much torque (or force or twist) to apply
3. It does not matter which way the pole swings, left or right, as long as the pole balances.
4. Your task: You will see two clips of behaviors, and your job is to select the clip you think is better given the above objective. If you think both clips are identical in behavior, you can select equally preferable.

The second set of instructions (i.e., advice) for the study are as follows:

1. The first priority is for the pole to begin swinging back and forth (i.e., picking up momentum). Therefore, the video clip where the pole has swung higher is better. Even if the agent is not behaving well in either clips, if you can tell that the pole is higher in one clip than the other, it is better to prefer that clip.
2. In general, if the pole is swinging rapidly in a circle (e.g., complete 360), then this is usually worse behavior than if the pole is barely moving.

**Cartpole-swingup**    The first set of instructions for the study are as follows:

1. Pole is attached to a cart.
2. Agent can move the cart left and right.
3. Goal is for the agent to move the cart such that the pole swings up and balances
4. Your task: You will see two clips of behaviors, and your job is to select the clip you think is better given the above objective. If you think both clips are identical in behavior, you can select equally preferable.

The second set of instructions (i.e., advice) for the study are as follows:

1. The first priority is for the pole to begin swinging back and forth (i.e., picking up momentum). Therefore, the video clip where the pole has swung higher is better. Even if the agent is not behaving well in either clips, if you can tell that the pole is higher in one clip than the other, it is better to prefer that clip.
2. In general, if the pole is swinging rapidly in a circle (e.g., complete 360), then this is usually worse behavior than if the pole is barely moving.
3. If the pole is balancing then falls over, then is is better behavior than if the pole is barely moving.

During the instruction period, participants also watched video clips of what successful and unsuccessful behavior looks like. In addition, participants were able to ask questions about the environment's objective.

Once the instruction period was complete, participants completed a practice round in providing preferences. This was included so users could gain familiarity with the interface. Figure 7 shows a screenshot of the interface used. Participants would use keyboard input to move from one video clip to the next. Then, participants had the opportunity to rewatch either clips as many times as they liked. Afterwards, participants chose which clip they preferred (or equally preferred) via keyboard input. Each video clip consisted of a 50 step segment, which was approximately 2 seconds long. The total study time was $\sim (1 - 1.5)$ hours. After the study was complete, participants filled out a demographic survey which included questions pertaining to their age, race/ethnicity, education level completed, and area of expertise.

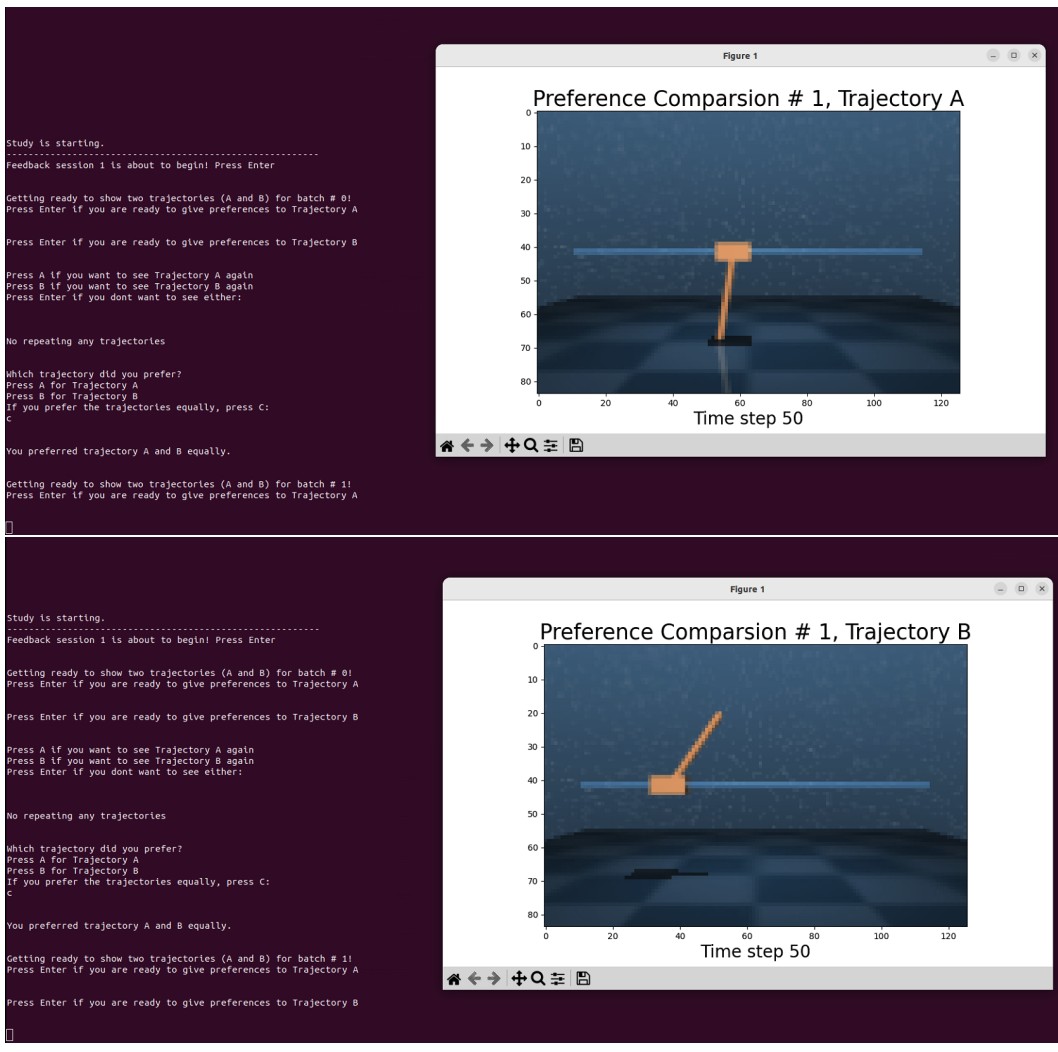

Figure 7: This shows the user interface used for the human subject study. Participants would view each video clip, sequentially, then decide which clip they preferred.

## B.2 PREFERENCE LEARNING SURVEY DETAILS

Table 6 provides a list of all papers included in our preference learning survey. We now describe the criteria for inclusion and exclusion in our survey. To find papers, we used Google Search with key words: preference learning, reinforcement learning from human feedback, and preference based reinforcement learning. We also found papers by reviewing those that cited popular preference learning works. This included Christiano et al. (2017) and Lee et al. (2021a), as these were two of the first papers that popularized preference learning. We included conference, journal and workshop papers in our survey. We did not include papers that were strictly on Arxiv. To keep the survey within scope, we did not include any preference learning works related to large language or foundation models. We also did not include any works that were entirely theoretical (e.g., no empirical results were provided).

Table 6: Articles used in Preference Learning Survey from Section 5.3. NA indicates no information was provided.

| PAPER REFERENCE | HUMAN USERS (NON-AUTHOR) | NUMBER OF USERS |
| --- | --- | --- |
| (Myers et al., 2023) | Yes | 22 |
| (Metcalf et al., 2024) | Yes | 50 |
| (Myers et al., 2022) | Yes | 50 |
| (Christiano et al., 2017) | Yes | NA |
| (Hejna III & Sadigh, 2023) | Yes | 4 |
| (Hwang et al., 2023b) | Yes | 5 |
| (Biyik & Sadigh, 2018) | Yes | 10 |
| (Hwang et al., 2023a) | Yes | 5 |
| (Bıyık et al., 2020) | Yes | 10 |
| (Wilde et al., 2021) | Yes | 18 |
| (Bıyık et al., 2022) | Yes | 15 |
| (Ibarz et al., 2018) | Yes | NA |
| (Knox et al., 2023) | Yes | 143 |
| (Holk et al., 2024) | Yes | 32 |
| (Metcalf et al., 2023) | Yes | 40 |
| (Marta et al., 2024) | Yes | NA |
| (Marta et al., 2023a) | Yes | 70 |
| (Marta et al., 2023b) | Yes | 20 |
| (Ren et al., 2022) | Yes | NA |
| (Mehta & Losey, 2024) | Yes | 15 |
| (White et al., 2024) | Yes | 20 |
| (Wang et al., 2022) | Yes | 10 |
| (Wang et al., 2023) | No | |
| (Zhang & Kashima, 2024) | No | |
| (Lee et al., 2021a) | No | |
| (Liang et al., 2022) | No | |
| (Park et al., 2022) | No | |
| (Hu et al., 2024) | No | |
| (Liu et al., 2022) | No | |
| (Liu et al., 2023) | No | |
| (Xue et al., 2024) | No | |
| (Daniels-Koch & Freedman, 2022) | No | |
| (Verma et al., 2023) | No | |
| (Barnett et al., 2023) | No | |
| (Verma & Metcalf, 2024) | No | |
| (Metcalf et al., 2023) | No | |
| (Swamy et al., 2024) | No | |
| (Lee et al., 2021b) | No | |
| (Wang et al., 2021) | No | |
| (Maxence Hussonnois & Rana, 2023) | No | |
| (Wilson et al., 2012) | No | |
| (Liu et al., 2024) | No | |
| (Giovanelli et al., 2024) | No | |
| (Zhu et al., 2024) | No | |
| (Cheng et al., 2024) | No | |

# C  ADDITIONAL HUMAN TEACHER RESULTS

| TASK | FEEDBACK | BACKGROUND | AUC | P VALUE | FINAL PERFORMANCE | P VALUE |
|------|----------|-----------|-----|---------|-------------------|---------|
| CARTPOLE-SWINGUP | 48 | Non-CS
CS | 8258.44 ± 1898.47
10208.11 ± 2311.16 | 0.183 | 648.88 ± 35.17
614.14 ± 114.81 | 0.692 |
| PENDULUM-SWINGUP | 40 | Non-CS
CS | 7677.84 ± 3540.70
10188.70 ± 3858.94 | 0.258 | 327.65 ± 407.80
641.68 ± 237.76 | 0.23 |

Table 7: This table shows the AUC and final performance of SDP (mean ± 95% confidence intervals) for CS and non-CS participants in the human subject study.

| Task | Feedback | Demographic | AUC | P Value |
|------|----------|-------------|-----|---------|
| CARTPOLE-SWINGUP | 48 | FEMALE
MALE | 8720.03 ± 3185.4
11770.22 ± 1669.96 | 0.085 |
| PENDULUM-SWINGUP | 40 | FEMALE
MALE | 8587.13 ± 2772.96
10134.67 ± 4822.22 | 0.331 |
| CARTPOLE-SWINGUP | 48 | AGE 18-30
AGE 50-70 | 10109.54 ± 2370.61
10923.05 ± 1292.83 | 0.313 |
| PENDULUM-SWINGUP | 40 | AGE 18-30
AGE 50-70 | 8619.07 ± 3381.48
10607.93 ± 5249.72 | 0.318 |
| CARTPOLE-SWINGUP | 48 | EDUCATION: BACHELORS OR HIGHER
EDUCATION: HIGH SCHOOL DIPLOMA | 11044.55 ± 1783.25
6247.99 ± 5186.47 | 0.794 |
| PENDULUM-SWINGUP | 40 | EDUCATION: BACHELORS OR HIGHER
EDUCATION: HIGH SCHOOL DIPLOMA | 10188.88 ± 3858.95
7677.85 ± 3540.7 | 0.742 |

Table 8: This table shows the AUC of SDP (mean ± 95% confidence intervals) for different demographic conditions. This table highlights that SDP can be effective irrespective of demographic background (gender, age, and educational background).

Tables 7-8 highlight that there is no significant difference in the performance of SDP when human teachers have differing backgrounds. This includes a background in CS, educational level, age, and gender. We note that we did not perform statistical analysis comparing racial groups as some experiments had only one participant of a particular racial background, making comparisons less meaningful.

# D    ADDITIONAL SIMULATED TEACHER RESULTS

For simplicity, in all additional experiments in this section, we only compare SDP + PEBBLE with PEBBLE (or R-PEBBLE).

## D.1    STUDY OF AGENT UPDATE PHASE

Figure 8 emphasizes how the agent update phase does result in new transitions, therefore the teacher provides feedback to transitions that are different from the original sub-optimal transitions used for pre-training.

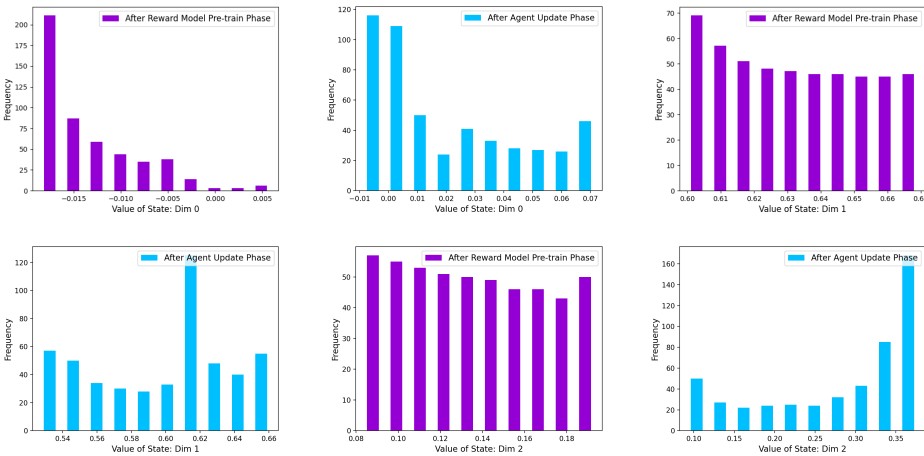

Figure 8: Door open, preference learning exp. These plots show how the agent's policy has changed from the reward model pre-training phase (purple histograms) to the agent update phase (blue histograms), thereby resulting in a different distribution for the state features.

## D.2    TRUE REWARD VALUES OF RANDOM POLICY

Figure 9 shows the true reward values for sub-optimal data gathered through a random policy in the DMControl suite. This emphasizes that the true reward value is close to the value we pseudo-label the sub-optimal transitions with (i.e., zero), therefore SDP should not yield a large incorrect reward bias.

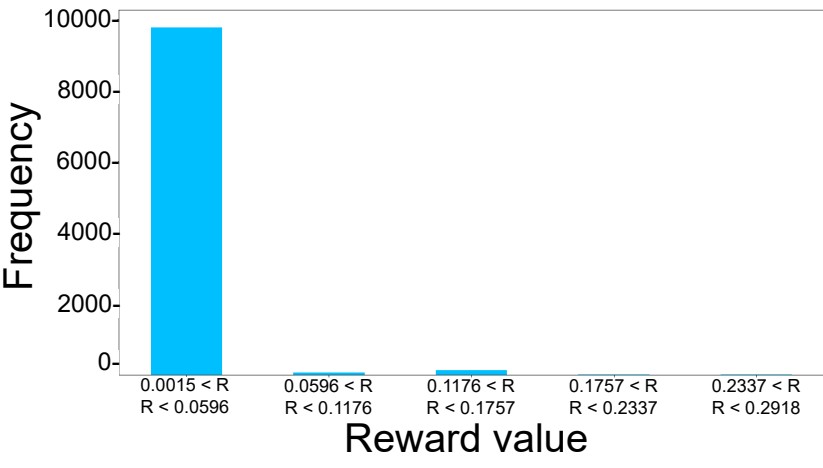

(a) Walker-walk.

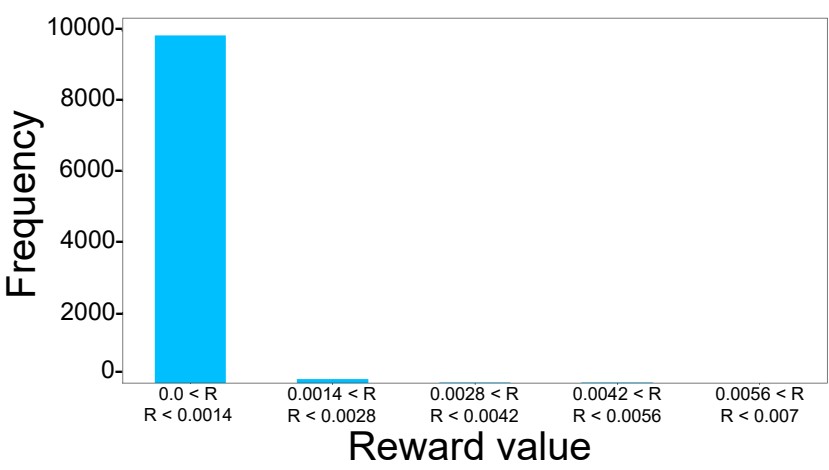

(b) Cheetah-run.

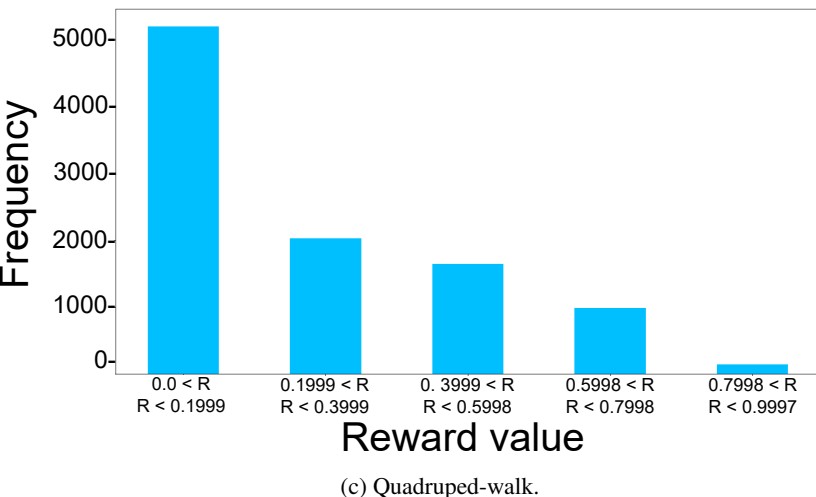

(c) Quadruped-walk.

Figure 9: Distribution of true reward values for transitions obtained with a random policy

### D.3 ZERO WEIGHT STUDY

To understand the effect of each component of SDP, we performed two additional ablations. First, in the reward model pre-training phase, the goal is for the reward model to learn to output zero. However, a trivial means to achieve an output of zero is to set all weights and biases in the neural network to zero. Therefore, we compare the full SDP to SDP using a zero-weight initialization as a replacement for the reward model pre-training. We found that using a zero-weight initialization for the reward model in place of the pre-training phase results in significantly degraded performance (see the green curve in Figure 10–right). This is not surprising, as previous works have found that a zero-weight initialization can negatively affect the training of neural networks (Blumenfeld et al., 2020; Zhao et al., 2022). In addition, Figure 11 demonstrates that the reward model pre-training phase does not produce reward model weights of zero, emphasizing why we do not experience the same performance degradation that occurs if we use a zero-weight initialization.

### D.4 FAKE INPUT STUDY

Second, SDP makes use of state, action transitions that are gathered through a sub-optimal policy. Therefore, these are transitions that an agent experiences while interacting with its environment. However, as previously noted, the goal of the reward model pre-training phase is for the reward model to learn to output zero. Therefore, is it necessary for the reward model to pre-train on inputs that are real environment transitions? Instead, can we pre-train the reward model on transitions that did not result from an agent-environment interaction? To test this, we created "fake" inputs of size dim(state) + dim(action), and for each input dimension, we randomly sampled a value from $\mathcal{N}(0, 1)$. We obtained 50,000 "fake" transitions and used this data for the reward model pre-training phase. In this experiment, our goal is to understand the effect of the type of inputs on the reward model pre-training phase, therefore we kept the agent update phase as is (i.e., provided the true sub-optimal data for this phase). We then compared SDP using true transitions to SDP using "fake" transitions in Figures 10–left and 12. We found that the full SDP (purple curve) in which we pre-train the reward model on true sub-optimal transitions yields significantly greater final performance than SDP using "fake" transitions (green curve). This highlights the importance of using true sub-optimal transitions in SDP.

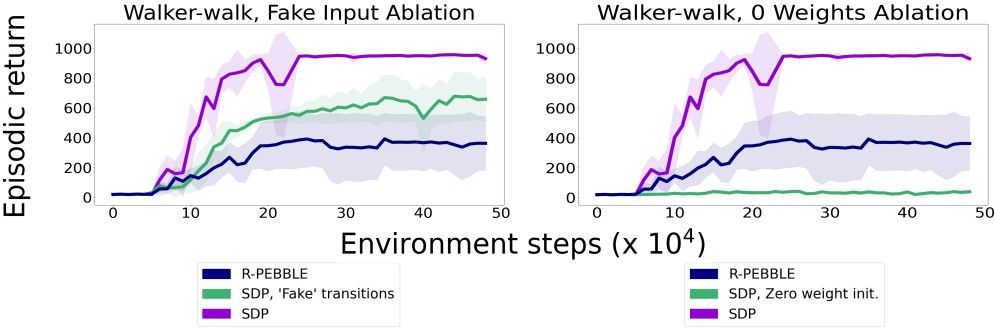

Figure 10: Zero weights and fake input studies in Walker-walk

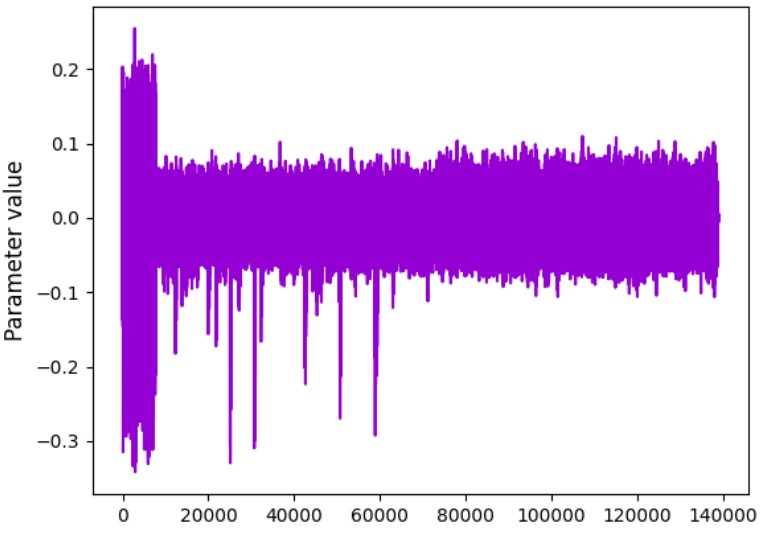

(a) Scalar feedback experiment in Walker-walk.

Figure 11: This figure shows the reward model weights of SDP after the reward model pre-training phase. This demonstrates that the reward model pre-train phase of SDP does not result in zero neural network weights.

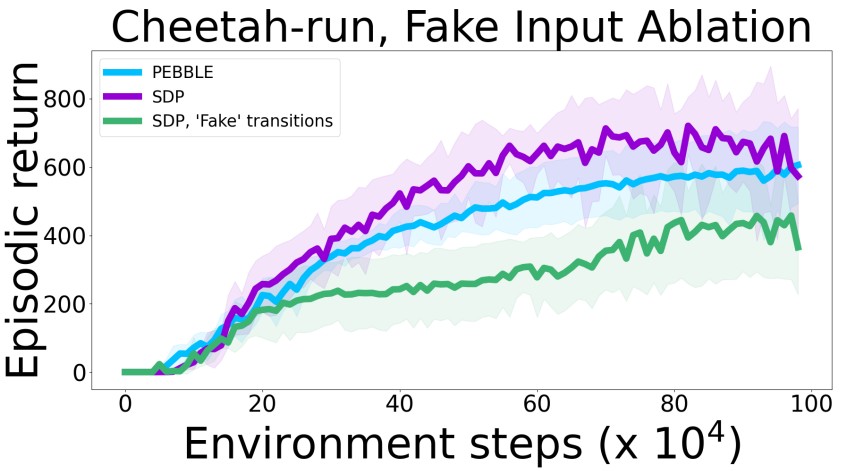

Figure 12: Fake Input Study in Cheetah Run Preference Learning

## D.5 SDP COMPONENT STUDIES

Figure 13 shows additional phase ablations in the preference learning experiments done on the Cheetah-run environment. This highlights the importance of using both phases in SDP. In addition, we ran an experiment in Walker-walk where we directly apply the procedure Yu et al. (2022) uses for leveraging sub-optimal data in the offline RL setting. They simply store the sub-optimal transitions in the RL agent's replay buffer with the pseudo reward label of 0 and follow standard offline RL. We repeated this with the other difference of following standard preference learning (PEBBLE) afterwards. We found that the average final performance was 104.32 with a 95% confidence interval of 46.12, which is close to 4 times less than the final performance of SDP.

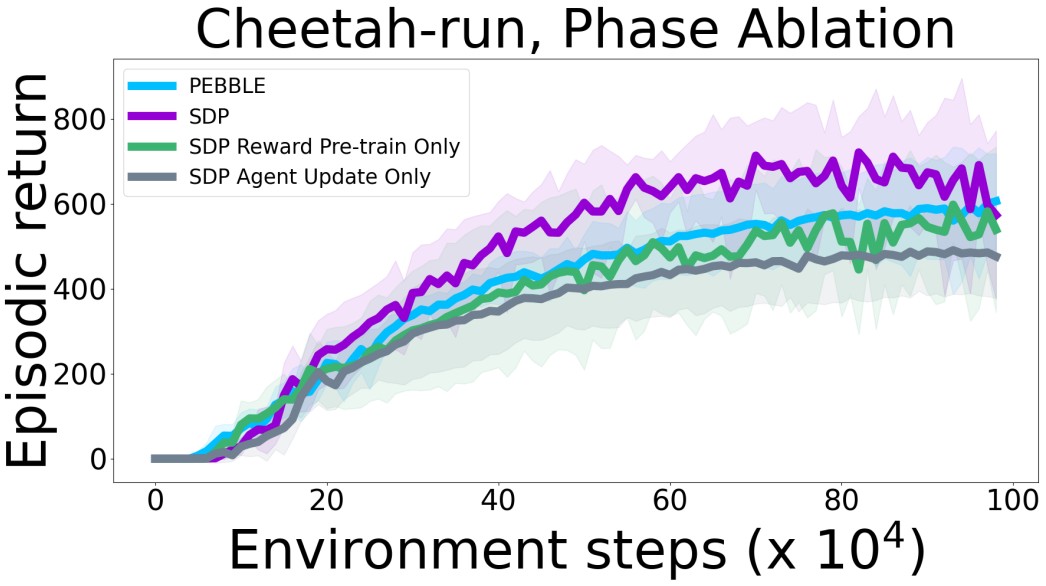

Figure 13: Phase Ablation Study in Cheetah Run Preference Learning

Figure 14 shows an additional ablation over the amount of prior data used in SDP. We observed similar performance gains as described in section 5.4.

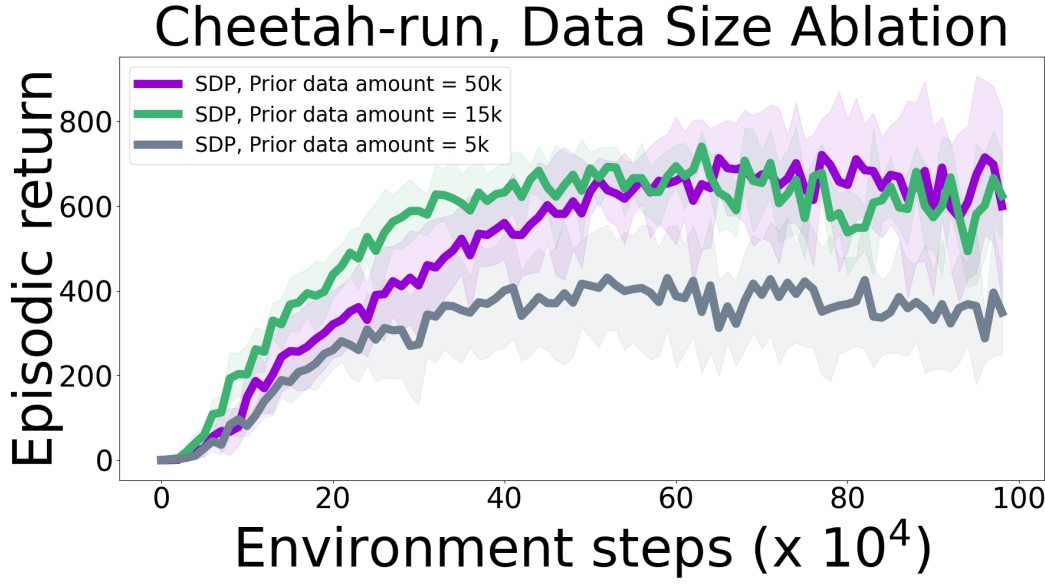

Figure 14: Prior Data Amount Study in Cheetah Run Preference Learning

## D.6 EFFECT OF RELABELLING REPLAY BUFFER

SDP is combined with four preference learning algorithms, PEBBLE, RUNE, SURF, and MRN. A core feature of these algorithms is that every time the reward model is updated, all transitions inside the RL agent's replay buffer are updated using the latest reward model. Figure 15 shows the effects of not relabeling the sub-optimal data (i.e., the reward labels) with the latest learned reward model. This means the reward labels remain frozen at zero throughout the entire training process. This ablation was to demonstrate that if the sub-optimal data in the agent's replay buffer is not updated with the latest reward model, then performance will suffer. This is likely the case because the incorrect reward bias from the pseudo-labeling process persists, whereas when we relabel the transitions with the updated reward model, the incorrect reward bias may reduce over time.

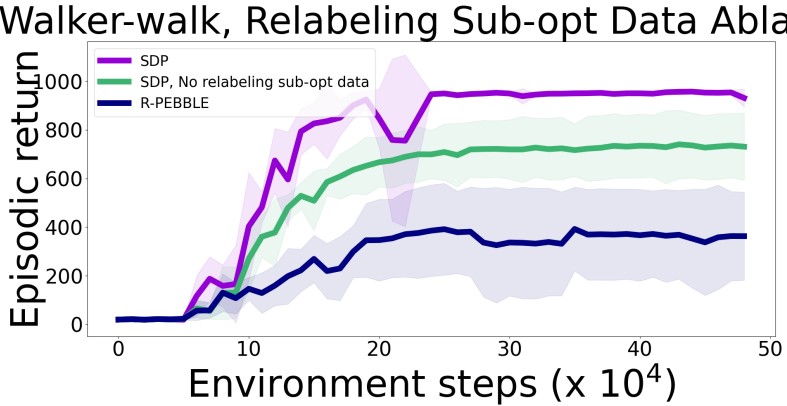

Figure 15: Relabeling sub-opt data study: Walker-walk, scalar feedback

## D.7 EFFECT OF TRANSITION QUALITY IN SDP

Moreover, we show that the effectiveness of SDP relies on the use of sub-optimal data transitions. If we use high-quality data transitions (i.e., transitions that came from a fully trained RL agent policy), SDP will fail (see Figure 16). This unsurprising result confirms that pseudo-labeling high-reward transitions with zero can significantly hurt the reward model and the agent's performance

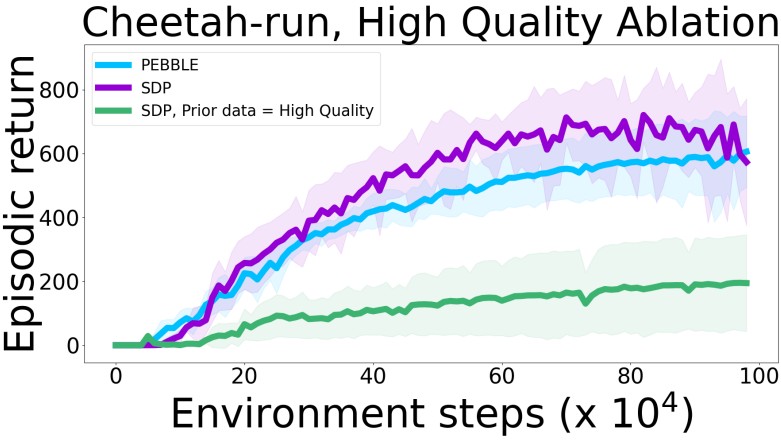

Figure 16: Using high-quality data in SDP study: Cheetah-run, preference feedback

## D.8 Scalar-based Experiment Statistics

Tables 9 and 10 provide a summary of the mean final performance and the mean area under the curve for all environments and benchmarks in the scalar feedback setting

| LEARNING FROM SCALAR FEEDBACK | | | |
|---|---|---|---|
| **Task** | **SDP** | **R-PEBBLE** | **DEEP TAMER** |
| WALKER-WALK | $931.31 \pm 36.59$ | $362.99 \pm 181.23$* | $33.10 \pm 11.10$* |
| CHEETAH-RUN | $862.72 \pm 49.13$ | $522.10 \pm 238.87$* | $30.71 \pm 16.26$* |
| QUADRUPED-WALK | $777.38 \pm 156.74$ | $543.92 \pm 193.21$* | $73.74 \pm 48.38$* |

Table 9: This table shows the mean final performance plus and minus the 95% confidence interval. * indicates SDP achieves a significantly greater mean final performance.

| LEARNING FROM SCALAR FEEDBACK | | | |
|---|---|---|---|
| **Task** | **SDP** | **R-PEBBLE** | **DEEP TAMER** |
| WALKER-WALK | $34981.65 \pm 1559.24$ | $13147.1 \pm 6102.31$* | $1823.42 \pm 578.48$* |
| CHEETAH-RUN | $55144.89 \pm 4266.94$ | $33557.58 \pm 15185.47$* | $2459.23 \pm 1226.34$* |
| QUADRUPED-WALK | $58370.85 \pm 10759.00$ | $37076.52 \pm 11968.48$* | $7365.68 \pm 3556.27$* |

Table 10: This table shows the mean area under the learning curve (AUC) plus and minus the 95% confidence interval. * indicates SDP achieves a significantly greater AUC.

## D.9 PREFERENCE LEARNING EXPERIMENT STATISTICS

Tables 11-14 provide a summary of the mean final performance and the mean area under the curve for all environments and benchmarks in the preference feedback setting.

| TASK | FEEDBACK | METHOD | FINAL RETURN | P VALUE |
|---|---|---|---|---|
| **CARTPOLE-SWINGUP** | **48** | PEBBLE | 226.4 ± 136.35 | **0.007*** |
| | | SDP + PEBBLE | 533.94 ± 57.71 | |
| | | RUNE | 263.48 ± 176.2 | |
| | | SDP + RUNE | 561.77 ± 108.17 | **0.021*** |
| | | SURF | 83.12 ± 44.53 | |
| | | SDP + SURF | 324.19 ± 179.8 | **0.039*** |
| | | MRN | 151.05 ± 76.31 | |
| | | SDP + MRN | 644.03 ± 57.99 | **0.0*** |
| **WALKER-WALK** | **100** | PEBBLE | 173.9 ± 100.26 | **0.001*** |
| | | SDP + PEBBLE | 490.07 ± 50.07 | |
| | | RUNE | 160.6 ± 61.85 | **0.0*** |
| | | SDP + RUNE | 575.71 ± 104.52 | |
| | | SURF | 202.1 ± 95.17 | **0.005*** |
| | | SDP + SURF | 435.99 ± 39.45 | |
| | | MRN | 215.14 ± 58.09 | **0.0*** |
| | | SDP + MRN | 672.3 ± 85.53 | |
| **CHEETAH-RUN** | **200** | PEBBLE | 582.51 ± 67.62 | **0.062** |
| | | SDP + PEBBLE | 704.63 ± 102.27 | |
| | | RUNE | 681.59 ± 68.05 | **0.645** |
| | | SDP + RUNE | 616.78 ± 279.34 | |
| | | SURF | 642.52 ± 84.3 | **0.266** |
| | | SDP + SURF | 682.34 ± 65.08 | |
| | | MRN | 728.36 ± 38.69 | **0.305** |
| | | SDP + MRN | 748.99 ± 55.59 | |
| **QUADRUPED-WALK** | **500** | PEBBLE | 350.24 ± 228.91 | **0.017*** |
| | | SDP + PEBBLE | 753.18 ± 104.79 | |
| | | RUNE | 363.65 ± 163.93 | **0.009*** |
| | | SDP + RUNE | 704.1 ± 102.16 | |
| | | SURF | 550.51 ± 211.19 | **0.089** |
| | | SDP + SURF | 767.78 ± 140.67 | |
| | | MRN | 200.12 ± 63.04 | **0.0*** |
| | | SDP + MRN | 733.83 ± 62.87 | |

Table 11: This table shows the final performance (mean ± 95% confidence intervals) for all DM-Control preference learning experiments. * indicates significant differences.

| TASK | FEEDBACK | METHOD | AUC | P VALUE |
|---|---|---|---|---|
| **CARTPOLE-SWINGUP** | 48 | PEBBLE | 8248.14 ± 2148.59 | **0.0*** |
| | | SDP + PEBBLE | 21519.9 ± 2435.02 | |
| | | RUNE | 8817.21 ± 3534.57 | **0.004*** |
| | | SDP + RUNE | 20970.54 ± 4643.96 | |
| | | SURF | 5593.44 ± 1099.56 | **0.074** |
| | | SDP + SURF | 10288.22 ± 4584.16 | |
| | | MRN | 6774.95 ± 1783.95 | **0.0*** |
| | | SDP + MRN | 22564.32 ± 3846.6 | |
| **WALKER-WALK** | 100 | PEBBLE | 7571.79 ± 2903.09 | **0.001*** |
| | | SDP + PEBBLE | 18458.81 ± 1224.69 | |
| | | RUNE | 6525.52 ± 2566.88 | **0.0*** |
| | | SDP + RUNE | 21059.76 ± 3056.28 | |
| | | SURF | 7675.24 ± 2857.99 | **0.001*** |
| | | SDP + SURF | 16700.51 ± 1449.62 | |
| | | MRN | 7711.05 ± 1295.46 | **0.0*** |
| | | SDP + MRN | 25184.26 ± 3071.65 | |
| **CHEETAH-RUN** | 200 | PEBBLE | 39245.94 ± 5189.16 | **0.006*** |
| | | SDP + PEBBLE | 51121.68 ± 3488.87 | |
| | | RUNE | 45824.63 ± 4459.04 | **0.792** |
| | | SDP + RUNE | 37202.31 ± 16330.83 | |
| | | SURF | 40522.18 ± 2989.22 | **0.218** |
| | | SDP + SURF | 42777.1 ± 3775.07 | |
| | | MRN | 47569.45 ± 2191.57 | **0.106** |
| | | SDP + MRN | 49968.24 ± 2187.97 | |
| **QUADRUPED-WALK** | 500 | PEBBLE | 20584.95 ± 9814.49 | **0.018*** |
| | | SDP + PEBBLE | 42044.34 ± 11183.84 | |
| | | RUNE | 22861.89 ± 9861.31 | **0.082** |
| | | SDP + RUNE | 32664.79 ± 3929.32 | |
| | | SURF | 30884.8 ± 11660.52 | **0.074** |
| | | SDP + SURF | 44176.44 ± 8404.76 | |
| | | MRN | 14805.07 ± 1860.39 | **0.002*** |
| | | SDP + MRN | 35209.32 ± 6225.09 | |

Table 12: This table shows the AUC (mean ± 95% confidence intervals) for all DMControl preference learning experiments. * indicates significant differences.

| TASK | FEEDBACK | METHOD | FINAL RETURN | P VALUE |
|---|---|---|---|---|
| **HAMMER** | **7500** | PEBBLE | 10.0 ± 13.58 | **0.003*** |
| | | SDP + PEBBLE | 66.0 ± 21.18 | |
| | | RUNE | 4.0 ± 7.01 | **0.001*** |
| | | SDP + RUNE | 68.0 ± 17.0 | |
| | | SURF | 0.0 ± 0.0 | **0.033*** |
| | | SDP + SURF | 36.0 ± 25.16 | |
| | | MRN | 4.0 ± 7.01 | **0.027*** |
| | | SDP + MRN | 46.0 ± 27.5 | |
| **DOOR-UNLOCK** | **500** | PEBBLE | 42.0 ± 34.35 | **0.066** |
| | | SDP + PEBBLE | 80.0 ± 15.68 | |
| | | RUNE | 26.0 ± 22.59 | **0.02*** |
| | | SDP + RUNE | 36.0 ± 38.65 | |
| | | SURF | 20.0 ± 35.06 | **0.354** |
| | | SDP + SURF | 80.0 ± 22.17 | |
| | | MRN | 48.0 ± 38.17 | **0.399** |
| | | SDP + MRN | 56.0 ± 37.02 | |
| **DOOR-LOCK** | **500** | PEBBLE | 38.0 ± 30.06 | **0.035*** |
| | | SDP + PEBBLE | 80.0 ± 7.84 | |
| | | RUNE | 62.0 ± 32.51 | **0.793** |
| | | SDP + RUNE | 40.0 ± 30.87 | |
| | | SURF | 70.0 ± 31.85 | **0.684** |
| | | SDP + SURF | 60.0 ± 13.58 | |
| | | MRN | 66.0 ± 26.93 | **0.268** |
| | | SDP + MRN | 78.0 ± 17.88 | |
| **DRAWER-OPEN** | **1000** | PEBBLE | 4.0 ± 7.01 | **0.045*** |
| | | SDP + PEBBLE | 36.0 ± 25.16 | |
| | | RUNE | 0.0 ± 0.0 | **0.001*** |
| | | SDP + RUNE | 66.0 ± 14.24 | |
| | | SURF | 20.0 ± 35.06 | **0.018*** |
| | | SDP + SURF | 80.0 ± 14.67 | |
| | | MRN | 20.0 ± 35.06 | **0.136** |
| | | SDP + MRN | 56.0 ± 40.21 | |
| **WINDOW-OPEN** | **200** | PEBBLE | 34.0 ± 37.44 | **0.234** |
| | | SDP + PEBBLE | 54.0 ± 26.35 | |
| | | RUNE | 12.0 ± 21.04 | **0.014*** |
| | | SDP + RUNE | 38.0 ± 24.42 | |
| | | SURF | 14.0 ± 20.44 | **0.098** |
| | | SDP + SURF | 66.0 ± 26.35 | |
| | | MRN | 46.0 ± 32.61 | **0.208** |
| | | SDP + MRN | 68.0 ± 31.06 | |

Table 13: This table shows the final performance (mean ± 95% confidence intervals) for all Meta-world preference learning experiments. **\*** indicates significant differences.

| Task | Feedback | Method | AUC | P Value |
|---|---|---|---|---|
| **Hammer** | **7500** | PEBBLE | 424.0 ± 175.18 | **0.017*** |
| | | SDP + PEBBLE | 2222.0 ± 1009.47 | |
| | | RUNE | 490.0 ± 300.85 | **0.001*** |
| | | SDP + RUNE | 2520.0 ± 627.47 | |
| | | SURF | 482.0 ± 215.59 | **0.006*** |
| | | SDP + SURF | 3042.0 ± 1073.54 | |
| | | MRN | 494.0 ± 311.76 | **0.016*** |
| | | SDP + MRN | 2494.0 ± 1108.52 | |
| **Door-unlock** | **500** | PEBBLE | 2142.0 ± 1738.06 | **0.031*** |
| | | SDP + PEBBLE | 4838.0 ± 1251.13 | |
| | | RUNE | 1272.0 ± 1275.71 | **0.047*** |
| | | SDP + RUNE | 1836.0 ± 2063.68 | |
| | | SURF | 1580.0 ± 2462.83 | **0.348** |
| | | SDP + SURF | 4802.0 ± 1537.81 | |
| | | MRN | 3082.0 ± 2232.43 | **0.342** |
| | | SDP + MRN | 3842.0 ± 2230.38 | |
| **Door-lock** | **500** | PEBBLE | 876.0 ± 647.87 | **0.008*** |
| | | SDP + PEBBLE | 2300.0 ± 135.0 | |
| | | RUNE | 1342.0 ± 590.05 | **0.809** |
| | | SDP + RUNE | 884.0 ± 633.87 | |
| | | SURF | 1892.0 ± 842.85 | **0.429** |
| | | SDP + SURF | 1986.0 ± 184.83 | |
| | | MRN | 1966.0 ± 751.51 | **0.306** |
| | | SDP + MRN | 2236.0 ± 479.9 | |
| **Drawer-open** | **1000** | PEBBLE | 314.0 ± 533.02 | **0.117** |
| | | SDP + PEBBLE | 1702.0 ± 1706.12 | |
| | | RUNE | 38.0 ± 49.71 | **0.004*** |
| | | SDP + RUNE | 3794.0 ± 1306.6 | |
| | | SURF | 696.0 ± 1167.88 | **0.015*** |
| | | SDP + SURF | 4078.0 ± 1839.07 | |
| | | MRN | 1550.0 ± 2690.98 | **0.211** |
| | | SDP + MRN | 3206.0 ± 2114.16 | |
| **Window-open** | **200** | PEBBLE | 794.0 ± 1015.04 | **0.131** |
| | | SDP + PEBBLE | 1698.0 ± 822.8 | |
| | | RUNE | 230.0 ± 334.14 | **0.01*** |
| | | SDP + RUNE | 948.0 ± 631.92 | |
| | | SURF | 242.0 ± 248.44 | **0.064** |
| | | SDP + SURF | 2048.0 ± 867.93 | |
| | | MRN | 1252.0 ± 796.8 | **0.223** |
| | | SDP + MRN | 1728.0 ± 666.48 | |

Table 14: This table shows the area under the curve (mean ± 95% confidence intervals) for all Metaworld preference learning experiments. * indicates significant differences.

