# OpenReview forum: "Leveraging Sub-Optimal Data for Human-in-the-Loop Reinforcement Learning"
_ICLR.cc/2025/Conference — ICLR 2025 Poster_

### Official Review · Reviewer_TrDy · 2024-11-03

**Soundness:** 3
**Presentation:** 4
**Contribution:** 3
**Rating:** 8
**Confidence:** 4

**Summary:**

This paper presents a method to initialize a reward network for RL with human feedback. Their method centers around using random exploration to then label sub-optimal trajectories with low rewards and therefore filter those state action pairs from exploration once the human preference phase begins. Their results show that this method is able to improve results in baselines and is agnostic to which human preference algorithm used.

**Strengths:**

Research Problem. The research problem is at a high level “how to use unlabeled data to warm start RL with human preferences”. Since human preferences (or even AI preferences) are a reasonable and increasingly common method for learning reward models as opposed to hand coding them improving such algorithms with cheap data is a good research problem.

Novelty. The novelty seems reasonable from the related work section.

Significance/Contribution. The significance of this paper is solid. It has a high practicality in many empirical applications and I am of the opinion that works that improve data efficiency are critical in RL research.

Algorithm Details. The algorithm itself isn’t extremely complex but the authors do a good job explaining it.

Experimental Selection. The experiments ran emphasize the point of the paper well.

Presentation of Results. All figures are very well done and very clear.

Ablation Studies. Sensitivity study is done with hyperparams and shows relative robustness.

Empirical Results. Good, results indicate this is a successful method and general strategy for improving preferences.

Failure Analysis. There is no failure analysis in this paper. I wish for the Rune + SDP baseline you would talk about why it doesn’t help in this case. Is there a particular part of rune that makes it work worse? In what future algorithms would this strategy not work.

Clarity is generally good. See weakness for improvements.

**Weaknesses:**

I think the related work section could be improved. I understand that this work is human in the loop RL but I feel the human in the loop RL section is lacking. None of this section is actually compared to this work and doesn’t emphasize what makes this work novel. I understand that it isn’t directly related to most of these, but then I believe you should choose papers that this work relates more directly to (improving reward models for RL). I would recommend you shorten the Human in the loop section and add a section based on improving reward models.

I have a few issues with the experimental section. You say over 5 seeds. Does that mean each bar is only the average/CI of 5 runs? If I just missed the number let me know, if I didn’t miss it and that is correct, I’d recommend running these many more times (I realize it might take a while but 5 isn’t enough). You say the that the addition of SDP never huts performance. Your plots in Figure 2 suggests otherwise. It generally improves performance but for rune SDP hurts performance in a few of the tasks. If you mean never hurts performance by a statistically significant margin this is fine but I would prefer you say this explicitly as this is slightly misleading in my opinion.

Is it possible to hypothesize why not only does this improve sample efficiency but from Figure 4 seems to also improve finals results?

Clarity: I wish the authors would more explicitly/intuitively define some of the terms. I was able to figure it out eventually but helping the reader would be great. Specifically:
-	“Low quality transitions”
-	“Minimum environment reward”
-	“Feedback efficiency”
I wish the authors didn’t use the term “RQ”. Instead, if it possible, redefine them in the paragraph where you discuss them. No reason to make the reader jump around.

**Questions:**

There are no baseline comparison here. Is this the only work that you could augment the reward function with? I know there are other methods which do data augmentation possibly for the reward function, is there no such reasonable comparisons? If no that is fine but I would imagine there is (maybe not limit yourself to only pre-trained options if needed?).

How do we determine r_min?

What future work is there?

---

> ### Author Response · Authors · 2024-11-20
> **Response # 1 to Reviewer TrDy**
>
> Weakness # 1:
> - In the Human-in-the-Loop section of the Related Work, we briefly describe the five baselines used in our Experiments section, which include the preference learning algorithms: PEBBLE, SURF, RUNE, MRN, and the scalar feedback algorithm, Deep TAMER. To improve clarity, we will refer to these algorithms by name throughout this section. Additionally, we will emphasize that, like our method SDP, all the preference learning algorithms also learn a reward model, making this aspect more explicit in the revised section.
> - We would also be happy to include additional relevant related work if you were able to make some suggestions? We think we have covered the most relevant works but are always interested in identifying things we’ve missed.
>
> Weakness # 2:
> - We will revise the submission to make it explicit that SDP never hurts performance by a statistically significant margin.
>
> Question # 1:
> - We augmented SDP with four preference learning algorithms: PEBBLE, RUNE, SURF, and MRN. Of these, SURF incorporates data augmentation within its algorithm. We compared the performance of the baseline algorithm both with and without the addition of SDP. These algorithms were chosen as baselines for two reasons: (1) they are popular preference learning methods in the literature, and (2) both these baselines and our work aim to improve the feedback efficiency of preference learning. In other words, the goal is to develop a preference learning algorithm that can learn an accurate reward model with fewer human queries.
>
> Question # 2:
> - SDP does assume access to the minimum environment reward (r_min). However, we note that this assumption is commonly made in previous works [1,2]. However, future work can consider ways to relax this assumption. For example, one possibility is to use the lower bound of the reward model’s output as r_min.
>
> Question #3:
> - In future work, we plan to provide a theoretical analysis of SDP. Specifically, in the context of learning from scalar feedback, where a reward model is learned through regression. We intend to demonstrate that the reward bias introduced by SDP does not outweigh the reduction in model variance. Additionally, we intend to evaluate the efficacy of SDP with human teachers in more complex environments, such as a real robot performing a pick-and-place task. This is crucial for further validating the applicability of SDP to real-world scenarios. These points will be added to Section 6 of our submission.
>
> References:
> - [1] https://arxiv.org/abs/2202.01741
> - [2] https://arxiv.org/abs/2010.14500

---

### Official Review · Reviewer_85xh · 2024-11-03

**Soundness:** 3
**Presentation:** 3
**Contribution:** 3
**Rating:** 6
**Confidence:** 4

**Summary:**

The paper introduces Sub-optimal Data Pre-training (SDP), a method for improving human-in-the-loop reinforcement learning by pre-training reward models with sub-optimal, reward-free data to enhance feedback efficiency and performance.

**Strengths:**

1. This SDP approch is novel
2. This paper has some real world validation (human study)

**Weaknesses:**

1. The performance of the SDP method is highly dependent on the quality and representativeness of the sub-optimal data used for pre-training. Limited data availability or random policies generated from the same initial seeds can negatively impact SDP performance.
2. The human study in this paper did not account for human variance. For instance, would differences in education levels affect the quality of the labels?

**Questions:**

1. Is learning with the SDP method more effective than learning with sparse rewards?
2. Does SDP require more data for more complex environment?
3. How well does the SDP approach perform when applied to high-dimensional action or state spaces, such as humanoid?
4. Does the human teacher need to undergo training?

---

> ### Author Response · Authors · 2024-11-20
> **Author Response #1 to Reviewer 85xh**
>
> Weakness # 2:
> - We have included a comparison of the effectiveness of SDP for users with and without a computer science background and found that there was no significant difference (Table 7, Appendix C).
> - We have performed additional statistical testing comparing the effectiveness of SDP across the following demographic conditions: gender, age, and educational background. We found no significant differences across these dimensions. We have updated the manuscript with this result in Table 8, Appendix C.
> | Task              | Feedback | Demographic                      | AUC                 | P Value |
> |--------------------|----------|----------------------------------|---------------------|---------|
> | Cartpole-swingup  | 48       | Female                           | 8720.03 ± 3185.4    | 0.085   |
> |  Cartpole-swingup  | 48        | Male                             | 11770.22 ± 1669.96  |
> |Pendulum-swingup  | 40       | Female                           | 8587.13 ± 2772.96   | 0.331
> |Pendulum-swingup  | 40        | Male                             | 10134.67 ± 4822.22  |         |
> | Cartpole-swingup  | 48       | Age 18-30                        | 10109.54 ± 2370.61  | 0.313   |
> |  Cartpole-swingup  | 48        |Age 50-70                        | 10923.05 ± 1292.83  |         |
> | Pendulum-swingup  | 48       | Age 18-30                        |  8619.07 ± 3381.48   | 0.318   |
> |  Pendulum-swingup  | 48        |Age 50-70                        | 10607.93 ± 5249.72  |         |
> | Cartpole-swingup  | 48       | Education: Bachelors or higher   | 11044.55 ± 1783.25  | 0.794   |
> |  Cartpole-swingup  | 48        |Education: High school diploma   | 6247.99 ± 5186.47    |         |
> | Pendulum-swingup  | 48       | Education: Bachelors or higher   | 10188.88 ± 3858.95  | 0.742   |
> |  Pendulum-swingup  | 48        |Education: High school diploma   | 7677.85 ± 3540.7|         |
>
> Question # 1:
> - Yes, SDP can be more effective than traditional RL methods that learn from sparse rewards. Importantly, SDP learns a dense reward function from human feedback. RL algorithms historically have more difficulty learning in sparse reward settings for a variety of reasons: (1) credit assignment problem – when the agent receives a reward, it must figure out which states and actions it should credit for the reward, (2) slower training – the agent can only update its actor/critic when it receives a reward which infrequently occurs in the sparse setting. SDP, by contrast, can provide an intermediate reward, helping the agent learn more effectively.
>
> Question # 2:
> - In our experiments, we found that more complex environments (e.g., Quadruped-walk) performed well with the same amount of prior data as less complex environments (e.g., Cartpole-swingup). However, it is likely the case that in even higher dimensional environments, more sub-optimal data would be useful in order to achieve a better coverage of the sub-optimal state and action space.
>
> Question # 3:
> - We have evaluated SDP on both Quadruped-walk (main result – Figure 2) and Quadruped-run (ablation – Figure 4), each of which has an observation size of 78, whereas Humanoid from the DMControl suite has an observation size of 67. We observed that in Quadruped-walk, the addition of SDP was able to statistically outperform MRN and PEBBLE, and achieved comparable performance to SURF and RUNE. Similarly, in Figure 4, we found that SDP achieved greater learning efficiency than PEBBLE in Quadruped-run.
>
> Question # 4:
> - For any preference learning method, it is crucial that human teachers understand the objective of the task. To facilitate this, we provided detailed instructions outlining the task’s goal in our human-subject study. Additionally, we included video clips illustrating both task success and failure to ensure clarity. Familiarity with the preference collection interface also plays an important role in reducing errors. To address this, participants were given a practice round of providing preferences. These measures help ensure that the collected preferences are accurate and reliable. In Appendix B.1, we outlined in detail the protocol of our human-subject study.

---

> > ### Comment · Reviewer_85xh · 2024-11-27
> >
> > Appreciate the rebuttal. I think the changes made improve the paper.

---

### Official Review · Reviewer_xKys · 2024-11-04

**Soundness:** 3
**Presentation:** 4
**Contribution:** 2
**Rating:** 6
**Confidence:** 3

**Summary:**

Use reward-free sub-optimal data for pretraining human in the loop RL by adding a reward. The reward is trained by assigning the minimum environment reward to all states in the reward free suboptimal data, and iteratively adding interactions. The online interactions can be labeled either with preference-based RL or with scalar rewards. This suboptimal-data aware method is then evaluated over several experiments against other human in the loop algorithms.

**Strengths:**

The algorithm is straighforward.

The method is well explained and clear.

The empirical analysis is good.

Running human in the loop with human participants is admirable.

**Weaknesses:**

The assumption made in Equation 4 is quite restritive, since this states that the per-state reward must be low for every state in a suboptimal trajectory. There are many tasks for which a sub-optimal trajectory may receive significant non-trivial reward, suhc as achieving an intermediate goal, while still being significantly suboptimal. In addition, knowledge of the minimum environment reward can also be a limiting assumption, though less so.

The method may be a little bit too straightforward, as the concept of applying a small, fixed, low reward given to unlabeled interaction data has been explored in the offline/off-policy RL literature, which has a lot of overlap with human in the loop RL. Since the method proposes very little to convert this interaction data into the context of human in the loop, this makes the algorithm itself of low novelty. It seems like knowledge about the scale or type of human in the loop data (preference or scalar) would provide some insight into a better method for assigning the suboptimal data.

How does relabeling the suboptimal data with the preference learned rewards through training compare? This seems like a logical experiment that could provide insight into how much the suboptimality assumption can be relaxed.

While the baseline evaluation is extensive, it seems like a missing area are methods that utilize suboptimal data. As a comparison, pretraining with an offline RL method that can leverage suboptimal data smoothly would be preferred, at least as an upper bound on performance.

While user load is a serious concern, a participant study that investigated more complex domains, and experiments in more complex domains such as Franka Kitchen or 2D minecraft would illustrate the efficacy of this method. In these control tasks it is not particularly realistic to have a large amount of entirely reward-free data, whereas in a robotic kitchen this is more plasible. Since this method is fundamental, an experiment along this line would make sense.

**Questions:**

See Weaknesses

---

> ### Author Response · Authors · 2024-11-20
> **Author Response #1 to Reviewer xKys**
>
> Weakness # 2:
> - One of the key strengths of SDP is its simplicity, which offers several advantages. Simpler algorithms are often easier to implement, debug, and maintain which can facilitate broader adoption.
> - In addition, the use of sub-optimal data has been explored in the offline RL literature [1]. However, we note that in Appendix D.5, we included an experiment where we directly adapted this technique to the preference learning setting in the Walker-walk environment. Our results showed that the average final performance achieved using this adapted approach was nearly 4 times lower than the final performance of SDP. In addition, in Section 5.2, Figure 6 we performed an ablation of the components of SDP, the reward model pre-training phase, and the agent update phase. This study demonstrated that neither component, when used individually, leads to improved performance. Together, these experiments highlight that naively incorporating suboptimal data does not yield performance gains in the human-in-the-loop RL setting.
>
> Weakness # 3:
> - We conducted an experiment in the Cartpole-Swingup environment where we replaced the pseudo-labels of r_min​ with the labels generated by the preference-learned reward model. However, this ablation significantly underperformed compared to SDP. The primary issue with this ablation is that SDP relies on the reward model pre-training phase, which occurs before any human feedback is provided. In this ablation, however, the suboptimal data used to pre-train the reward model are pseudo-labeled with random rewards. These rewards are random because the preference-learned reward model has not yet been updated with human feedback. As a result, its outputs are random and fail to produce meaningful pseudo-labels, negating the usual benefits of the reward model pre-training phase.
> | Method                | AUC |
> |---------------------|------------------------|
> | SDP      | 21519.9 ± 2435.02                     |
> | SDP without r_min       | 7179.41 ± 505.97                  |
>
> Weakness # 4:
> - While we agree that evaluating against methods utilizing suboptimal data could be interesting, we believe that comparing to an offline RL algorithm would not be appropriate due to the vastly different problem settings and assumptions. Our work focuses on human-in-the-loop algorithms, where reward models are learned in real time through scalar or preference feedback, requiring both human input and online interaction with the environment. In contrast, offline RL assumes access to a pre-collected dataset of transitions spanning from low- to high-quality policies and may not allow for additional online interaction with the environment. These fundamental differences make direct comparisons between the two approaches challenging and less meaningful.
> - For an upper bound comparison, we argue that a better alternative is to consider the online RL algorithm SAC which learns directly from the ground truth reward function. This is a more appropriate benchmark for several reasons: (1) in our experiments in Section 5.2, the human-in-the-loop algorithms aim to approximate the ground truth reward function using preference or scalar feedback; (2) the human-in-the-loop algorithms we evaluate, including ours, use SAC as the base algorithm, ensuring consistency in comparison; and (3) as an online RL algorithm, SAC is expected to achieve higher performance than offline RL methods, which are inherently limited by the quality and diversity of the data they rely on.
> - We have included the performance of SAC in all environments to provide a clear upper bound for evaluating our approach and the baselines. That said, if all reviewers agree that this would be an important benchmark, we are open to adding such results to our revised submission.
>
> Weakness # 5:
> - We agree that investigating more complex domains would be valuable for illustrating the efficacy of our method in more realistic and challenging environments, and we plan to explore this in future work. However, in this submission, we specifically chose Cartpole-Swingup and Pendulum-Swingup to ensure that we could gather a larger sample size of participants. This decision allowed us to conduct a more robust evaluation of SDP in a controlled setting, where we could assess its performance across a broader range of users.
>
> References:
> - [1] https://arxiv.org/abs/2202.01741

---

> > ### Comment · Reviewer_xKys · 2024-11-25
> > **Response to Authors**
> >
> > I appreciate the new experiments and I believe the paper is improved, though many of the limitations remain. I believe the paper should be accepted and will maintain my score

---

### Official Review · Reviewer_RKAm · 2024-11-04

**Soundness:** 3
**Presentation:** 4
**Contribution:** 1
**Rating:** 6
**Confidence:** 4

**Summary:**

The submission proposed a simple framework, namely Sub-optimal Data Pre-training (SDP), that leverages reward-free, suboptimal data to warm-start the reward model as well as the RL agent replay buffer. Through simulation experiments with both simulated teacher policy and real human subjects, the authors claim that the proposed SDP can significantly improves the efficiency of human feedbacks under Human-in-the-loop (HitL) reinforcement learning setting.

**Strengths:**

- The main idea of this submission is straightforward and intuitive. Utilizing sub-optimal data is a popular way to improve sample efficiency.
- The submission is well-written and easy to follow. Visualizations are clear and helpful.
- The demonstrated experiments are reasonably comprehensive, invluding robotic locomotion and manipulation tasks.
- The submission includes a 16-people human subject study.
- The result analysis is well-formulated with multiple seeds and significant tests.

**Weaknesses:**

- My main concern about the submission is that the contribution is incremental without significant advance. Leveraging a set of sub-optimal data generated with randomly initialized policy as a warm-start for the reward model is straight forward, and seems to me the only major contribution. The authors follow the standard RLHF framework such as Bradley-Terry model, etc. This work failed to address any of the existing challenges, such as the assumption of linear reward feature combinations, the assumption of Boltzmann rationality of the human demonstrator, etc.

- I remain somewhat unconvinced in the claim made in line 205 - line 210. Pseudo-labeling all low-quality transitions with the same minimum environmental rewards indeed introduces bias into the reward model, and will further lead to reward ambiguity. The empirical results in the DMControl suite is not sufficient to claim that the bias for using an incorrect reward is low. In fact, I doubt the effectiveness in pseudo-labeling low-quality samples in more complex real-world robotic tasks with higher dimensions. It would be helpful if the authors can include additional experiments on higher-dimensional simulation or real-world experiments to showcase the effectiveness.

- The authors took a great effort to include a well-formulated human subject study. I don't think it deserves any extra credits as the human subjects are only asked to compare two clips and provide preference labels, instead of provide direct interventions (learning from human intervention). I think the user study with diverse human subjects will be mostly useful if they can provide direct intervention and correction trajectories. A single preference label over two trajectory pairs might not fully reflect human's intention.

**Questions:**

- Can authors provide more thorough explanation on why introduced bias into the reward model will not affect the performance?
- Can authors explain why Deep Tamer is not learning at all in Figure 3?
- In Figure 6 Phase Ablation, what is the difference between SDP Agent Update Only and R-PEBBLE? Why SDP Reward Pre-train Only has better performance over SDP Agent Update Only? A more detailed explanation on the difference between these baselines will be helpful.

---

> ### Author Response · Authors · 2024-11-20
> **Author Response #1 to Reviewer RKAm**
>
> Weakness # 1:
>
> - The primary focus of this work is on improving the feedback efficiency of human-in-the-loop reinforcement learning (HitL RL algorithms). While it is true that the work builds on established techniques, such as the Bradley-Terry model, our approach addresses the specific challenge of how to make more efficient use of limited human feedback (scalar ratings or preferences over trajectories). This is an open problem in the HitL literature and has significant practical implications, as common HitL RL techniques often require a large number (up to tens of thousands) of human queries, making it costly and labor-intensive.
>
> Question #1 / Weakness # 2:
>
> - It is important to emphasize that the effectiveness of SDP relies on the transitions being pseudo-labeled to be sub-optimal. In Equation (4), we define sub-optimal transitions as those for which the difference between the true reward and the minimum reward is smaller than a small threshold, epsilon. If this assumption holds, the bias introduced by using an incorrect reward label will be minimal, and the reward model will benefit from the larger dataset, leading to a reduction in model variance. However, if this assumption does not hold, SDP can negatively impact performance. In Appendix D.7 (Figure 16), we presented an experiment where we replaced sub-optimal transitions with those from an expert policy. In this experiment, we observed a performance degradation, highlighting that when the reward bias is large—such as when using expert policy transitions—SDP fails to improve performance.
>
> - Moreover, we evaluated the efficacy of SDP on two widely used testbeds for preference learning: the DMControl Suite and Metaworld, which are standard benchmarks in this domain [1-5]. Notably, 7 out of the 9 tested environments featured observation spaces exceeding 20 dimensions. In particular, we tested SDP on Quadruped-walk (main results) and Quadruped-run (ablation study), both of which have observation dimensions of 78. Future work could explore higher-dimensional settings, such as learning directly from pixel inputs.
>
> | Task                | Observation Space Size | Action Space Size |
> |---------------------|------------------------|-------------------|
> | Quadruped-walk      | 78                     | 12                |
> | Quadruped-run       | 78                     | 12                |
> | All Metaworld environments | 39              | 4                 |
> | Walker-walk         | 24                     | 6                 |
> | Cheetah-run         | 17                     | 6                 |
> | Cartpole-swingup    | 5                      | 1                 |
>
>
> Weakness # 3:
>
> - We thank the reviewer for recognizing the effort we put into the human subject study. We appreciate your suggestion regarding the potential benefits of including direct interventions and corrections in the user study. We agree that learning from human interventions, such as trajectory corrections, can provide more detailed insights into human intentions. However, this research focused on learning from feedback (scalar signals or preferences). Therefore, to test the efficacy of our method with real human teachers, requires us to perform a human-subject study where participants are providing preference-based feedback.
>
> - In general, preference-based feedback can be a more scalable and less labor-intensive method compared to direct interventions [6], and it has been shown to be effective in many real-world applications such as the rise of large language models. That said, we agree that future work should explore combining preference-based feedback with direct human interventions, which would allow for a richer understanding of human intent and could further improve the performance of preference learning algorithms.
>
> References:
> - [1] PEBBLE: https://arxiv.org/abs/2106.05091
> - [2] SURF https://arxiv.org/abs/2203.10050
> - [3] RUNE https://openreview.net/forum?id=OWZVD-l-ZrC
> - [4] MRN https://openreview.net/forum?id=OZKBReUF-wX
> - [5] Few shot: https://arxiv.org/abs/2212.03363
> - [6] https://users.cs.utah.edu/~dsbrown/readings/assist_teleop.pdf

---

> > ### Author Response · Authors · 2024-11-20
> > **Author Response #2 to Reviewer RKAm**
> >
> > Question # 2:
> >
> > - Deep Tamer uses scalar feedback to directly learn an action-value function through regression. This approach has a limitation: it only allows updates to the (state, action) pairs where feedback is provided. In other words, the model can only adjust its value estimates for those specific state-action pairs that receive scalar feedback during training. This can be restrictive because many (state, action) pairs may never receive feedback, as the amount of human feedback is limited. This can leave large portions of the model's (state, action) space under-updated. However, in settings where the feedback budget is not limited, Deep TAMER can be effective. We performed an additional experiment where we increased the feedback budget to 500000. Therefore, the Deep TAMER agent would receive feedback at every time step. In this setting, the performance of Deep TAMER improved.
> >
> > | Method                | AUC |
> > |---------------------|------------------------|
> > | Deep TAMER, budget = 5000      |1823.42 ± 578.48                 |
> > | Deep TAMER budget = 500000        | 35488.42 ± 412.16                    |
> >
> > - In contrast, SDP and R-PEBBLE learn a reward function from scalar feedback. The key advantage is that once the reward function is learned, it can be used to update any (state, action) pair the agent encounters, not just those with direct feedback.
> > - Additionally, the original Deep TAMER paper evaluated the algorithm on a single Atari game, demonstrating its ability to learn effectively in that context. This means that the evaluation may not have sufficiently tested the scalability or generalization of Deep TAMER’s approach across a broader set of tasks. Nonetheless, we emphasize that a comprehensive exploration of Deep TAMER’s use cases is beyond the scope of this paper
> >
> > Question # 3:
> > - SDP Agent Update Only is an ablation of the full SDP method in which the reward model is not pre-trained on the sub-optimal transitions. In this ablation, the sub-optimal transitions are stored only in the RL agent’s replay buffer. Afterward, the agent interacts with the environment and makes some number of learning updates before human feedback is queried. The purpose of the agent update phase is to allow the agent to generate new behaviors that a teacher can provide feedback on, specifically in comparison to the sub-optimal trajectories. This helps ensure that the teacher does not provide redundant feedback on the same sub-optimal transitions. However, this benefit is lost if the reward model pre-training phase is skipped. Without pre-training on sub-optimal transitions, the reward model lacks prior knowledge of these trajectories, therefore the teacher must provide feedback on these sub-optimal trajectories. This may explain why the SDP Agent Update Only variant underperforms the SDP Reward Model Pre-train Only variant. In the latter, the reward model still benefits from prior knowledge of the sub-optimal transitions, enabling it to have a useful reward initialization without requiring any human feedback.
> > - The primary difference between SDP Agent Update Only and R-PEBBLE is that in SDP Agent Update Only the RL agent’s replay buffer is initialized with the sub-optimal transitions with pseudo-labelled rewards. This may explain why the performance differences between SDP Agent Update Only and R-PEBBLE were not statistically different.

---

> > > ### Comment · Reviewer_RKAm · 2024-11-25
> > > **Response to Authors**
> > >
> > > I appreciate the additional experiments and that has address many of my concerns. I'll increase my score though there are still quite a bunch of limitations

---

### Author Response · Authors · 2024-11-20

We thank all reviewers for their comprehensive and thoughtful reviews. We have uploaded a revised version of the submission based on the reviewer's suggestions. All changes in the manuscript are highlighted in blue.

---

### Meta-Review · Area_Chair_3poJ · 2024-12-20

**Metareview:**

This paper presents a method for learning reward model from human by leveraging suboptimal data to warm-start the learning process. Minimal environmental rewards are assigned to reward free data and then the reward function is learned through scalar or preference feedback from human. Experiments in simulated domains with both oracle and human feedback show that the proposed method outperforms prior human-in-the-loop RL methods.

Reviewers find the proposed idea simple but novel, and appreciate the clarity of the presentation. Major concerns include 1) the assumptions made by the proposed algorithm can be limiting, 2) the complexity of current evaluation domains is relatively simple, and 3) the selection of baseline methods does not include offline RL algorithms.

Reviewers have reached a consensus to recommend acceptance after the discussion period.

**Additional Comments On Reviewer Discussion:**

The authors ran additional experiments during the discussion phase to help addressing concerns from the reviewers.

---

### Decision · Program_Chairs · 2025-01-22

Accept (Poster)